# Multinuclear 1D and 2D NMR with [19]F-Photo-CIDNP hyperpolarization in a microfluidic chip with untuned microcoil

M. Victoria Gomez [1,3] ✉, Sander Baas [2,3] & Aldrik H. Velders [1,2] ✉

Nuclear Magnetic Resonance (NMR) spectroscopy is a most powerful molecular characterization and quantification technique, yet two major persistent factors limit its more wide-spread applications: poor sensitivity, and intricate complex and expensive hardware required for sophisticated experiments. Here we show NMR with a single planar-spiral microcoil in an untuned circuit with hyperpolarization option and capability to execute complex experiments addressing simultaneously up to three different nuclides. A microfluidic NMR-chip in which the 25 nL detection volume can be efficiently illuminated with laser-diode light enhances the sensitivity by orders of magnitude via photochemically induced dynamic nuclear polarization (photo-CIDNP), allowing rapid detection of samples in the lower picomole range (normalized limit of detection at 600 MHz, $nLOD_{f,600}$, of 0.01 nmol $Hz^{1/2}$). The chip is equipped with a single planar microcoil operating in an untuned circuit that allows different Larmor frequencies to be addressed simultaneously, permitting advanced hetero-, di- and trinuclear, 1D and 2D NMR experiments. Here we show NMR chips with photo-CIDNP and broadband capabilities addressing two of the major limiting factors of NMR, by enhancing sensitivity as well as reducing cost and hardware complexity; the performance is compared to state-of-the-art instruments.

NMR, launched about eight decades ago[1], has become a widespread and indispensable analytical tool in a plethora of fields ranging from fundamental physics and chemistry to biology and medicine[2–5]. Hardware and software development are continuously improving NMR instrument performance and opening new areas of research. One of the more persistent bottlenecks still to be tackled concerns the intrinsic low sensitivity due to the low thermal polarization of nuclear spins, and as a result commercial systems have evolved into very complex and expensive pieces of hardware. The classical solution for improving sensitivity concerns increasing the Boltzmann factor by use of ever-stronger, viz. costly, magnets, currently $B_0$ of 28 T (1200 MHz 1H Larmor frequency)[6]. A second widely used, and

again complex and costly, approach to boost sensitivity regards the use of cryogenically-cooled radiofrequency (RF) coils and preamplifiers, allowing an additional 2–4× gain in sensitivity. State-of-the-art high-field systems combined with cryoprobes allow for sub-millimolar concentration sensitivity in 0.5 mL samples, corresponding to quantities of sample higher than nanomole amounts[7]. Here we present the combination of two alternative and complementary strategies to boost the NMR sensitivity: (1) using microcoils to increase the detector sensitivity and (2) hyperpolarizing the nuclear spin population to increase the NMR signal intensity. Moreover, we exploit the untuned planar spiral microcoils to simplify the RF transceiver circuitry and have broadband capabilities[8]. This setup

[1]IRICA, Department of Inorganic, Organic and Biochemistry, Faculty of Chemical Sciences and Technologies, Universidad de Castilla-La Mancha (UCLM), Av. Camilo José Cela 10, 13071 Ciudad Real, Spain. [2]Laboratory of BioNanoTechnology, Wageningen University, 6700 EK Wageningen, The Netherlands. [3]These authors contributed equally: M. Victoria Gomez, Sander Baas. ✉e-mail: mariavictoria.gomez@uclm.es; aldrik.velders@wur.nl

now concomitantly allows for sophisticated photo-CIDNP-hyperpolarized multinuclear NMR experiments.

The miniaturization of transceiver coils to sub-μL sample volumes, as proposed in 1995 by Sweedler and collaborators[9], with the archetypical micro-solenoid design, has opened up the field of small-volume NMR[10,11]. Microcoils have been investigated with various geometries, e.g., from solenoids[9], to helmholtz[12], planar spiral[13–15], stripline[16,17], and microslot[18]. Most of the microcoils are relatively cheap, and often can be designed, and in some cases even manufactured, in-house[19,20]. The cost-effectiveness and potential of home-built microcoils is well illustrated by the ESCARGOT microcoils we developed, with a simple copper wire embedded in a microfluidic PDMS device allowing high-resolution NMR at a (unit) cost of less than 1 Euro[21]. Moreover, miniaturization allows for efficient and tailor-made versatility in sample-handling, including microfluidic hyphenation to complementary analytical techniques, such as chromatography or mass spectrometry[10,22–24]. In 2014, we reported the concept of a home-designed single-planar microcoil for the application in multinuclear NMR experiments. In short, a microfluidic glass chip was equipped with a planar spiral microcoil covering a 25 nL sample volume, with the single coil connected to a high-band ($^1$H or $^{19}$F) RF- channel and a low-band (X-nuclide, i.e., $^{13}$C or $^{31}$P) channel, allowing homo- and heteronuclear 1D and 2D NMR experiments with a non-resonant single planar microcoil[8,25]. Understanding the underlying principles and potential use of broadband detectors is currently investigated[25–28].

Another strategy in sensitivity enhancement concerns hyperpolarization techniques, that aim to alter the thermal Boltzmann distribution of the spin states of nuclides, and in theory can increase sensitivity up to several orders of magnitude[29]. There are various approaches to hyperpolarization of solution-state NMR, but most variants use common, i.e., 5 mm, NMR tubes. The most widely studied hyperpolarization variant is Dynamic Nuclear Polarization (DNP), but limitations are the long polarization process, limited available time for NMR measurements, and the complexity and costs of the hardware[30,31].

Boero and coworkers elegantly combined the microwave- and RF-coils in a single-chip DNP design[32]. Alternative hyperpolarization strategies include, SABRE/Para-Hydrogen-Induced-Polarization[33–35] and Xenon pumping[36] based strategies. Photo-Chemically Induced Dynamic Nuclear Polarization, photo-CIDNP, was developed in the late 1960's[37,38], and later found applications in investigating, e.g., protein folding mechanism[39]. After decades with only a few specialized groups holding up and exploiting the technique[40,41], in recent years a rapidly growing community of users is arising[42–46]. Arguably the biggest bottleneck for more widespread use of photo-CIDNP lies in the rapid degradation of the required sensitizer molecules, and in the, related, practical limitations for optimizing the experimental parameters such as relative concentrations of the sensitizer and target components, and light power, among others. With the earlier mentioned cheap and home-built ESCARGOT PDMS-based NMR devices, we have introduced photo-CIDNP hyperpolarization on microliter-sized samples, with sub-picomole mass sensitivity[44]. The flow set-up allowed for efficient use of the photo-CIDNP concept, eliminating the principal drawback and limitation, i.e., the degradation of the photosensitizer. This set-up combines the best of worlds, with the fiber-optics efficiently illuminating the whole sample area whilst not being in contact with the sample, as is the case in most common photo-CIDNP set ups[47,48]. However, heteronuclear multidimensional experiments cannot be simply performed with a single, tuned solenoid. We now hypothesized that integrating untuned nanoliter NMR chips with photo-CIDNP capability would not only boost the NMR's mass and concentration sensitivity but also allow for multinuclear experiments. In particular, $^{19}$F is a 100% naturally abundant NMR active ($I = 1/2$) nuclide which finds many applications in supramolecular[49], (bio)materials[50], and medicinal chemistry[51], among others. Moreover, the $^{19}$F nuclides holds promise for photo-CIDNP experiments because of the often large electron-nucleus hyperfine coupling that determines hyperpolarization efficiencies.

In this work, we present a home-built design in which nanoliter-scale microfluidic NMR chips and photo-CIDNP hyperpolarization are

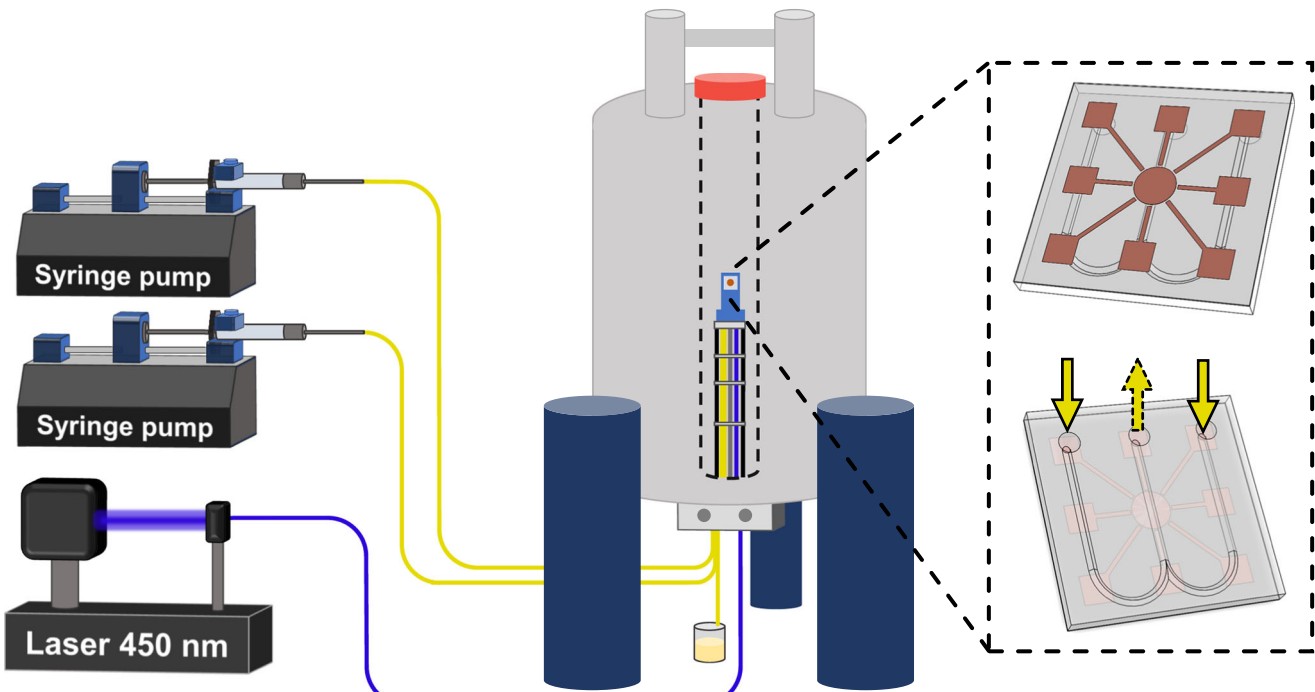

**Fig. 1 | Overview of the microfluidic photo-CIDNP NMR setup.** Two syringe pumps allow on-flow mixing and rapid optimization of photosensitizer/target ratio. The syringes are connected to the NMR chip via fused-silica capillaries. A third (outlet) capillary is connected to a waste container underneath the NMR magnet. A 450 nm fiber-coupled laser diode is used to illuminate in situ the sample volume under the planar spiral microcoil. On top of a probe base tube with RF transmission lines and CNC connectors at the bottom, a 3D-printed chip holder is mounted (blue), holding the chip in position (gray square).

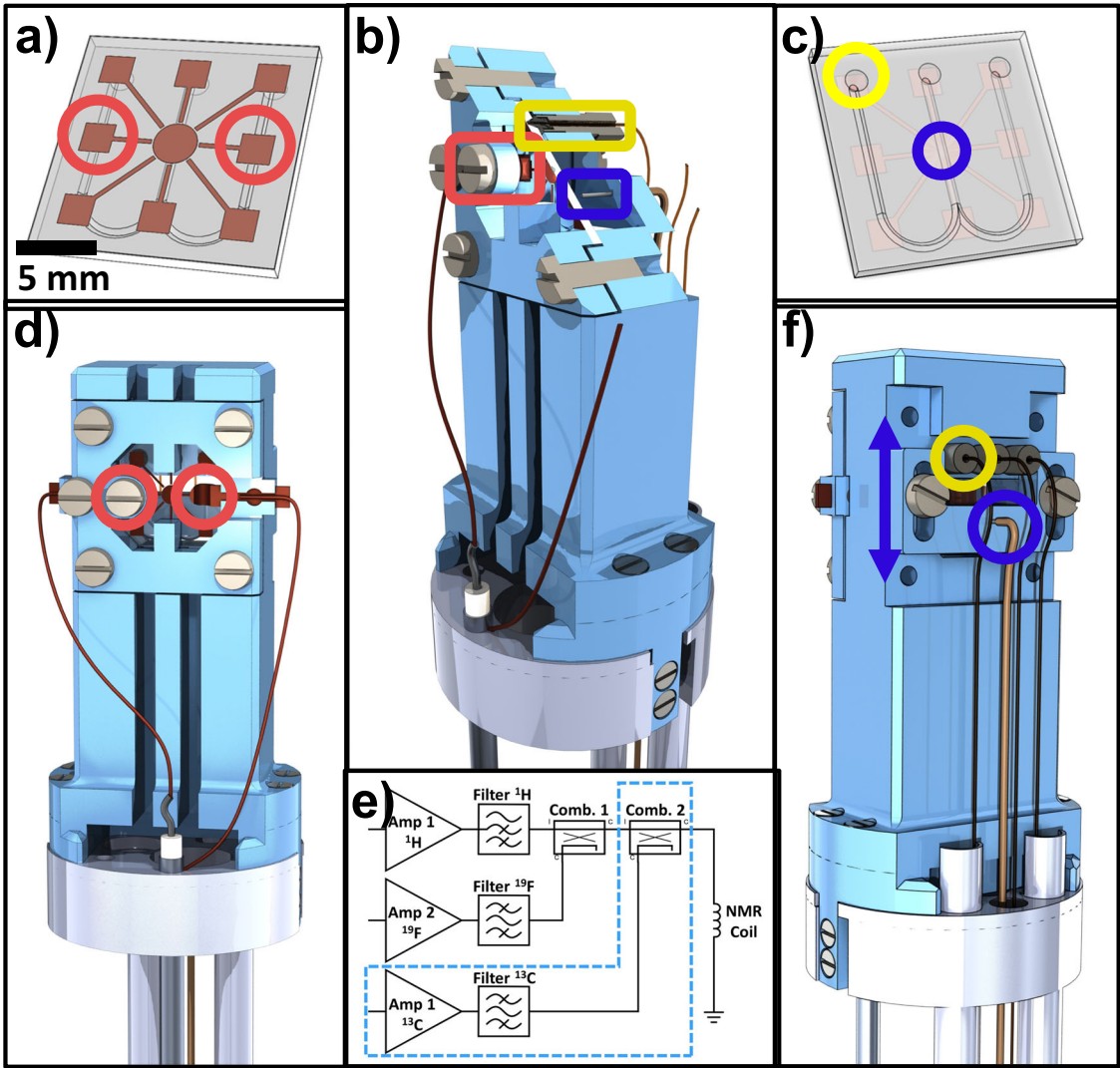

**Fig. 2 | Scheme of the NMR chip holder for planar spiral NMR-on-a-Chip.** Please note that the color coding of the highlighted features is chosen such to be consistent in the different figures: red for electrical, yellow for fluidic, and purple for optical fiber connections. **a** Top view of the microfluidic chip, where the planar spiral coil is displayed as a copper disc. Red circles indicate the contact pads which are connected to the microcoil central and outer conductor. **b** Detailed section view of the NMR chip holder. It consists of the main holder and a top clamp, which holds the chip in place. The ferrule system that connects the fused silica capillary to the microfluidic inlet/outlet is indicated by the yellow box. The red box indicates the M3 screw that pushes the copper contact clamp strip onto the NMR chip contact patch. The optical fiber bare end is directed toward the NMR chip (purple box). **c** Back view of the microfluidic NMR chip, showing the inlet/outlet (yellow circle). The sample can be illuminated from this side of the chip for Photo-CIDNP experiments (purple circle). **d** Front view of the NMR chip holder. Copper wires

connect the chip to the transmission line (left wire) and probe body ground (right wire). The outer, central screw pushes the copper wires onto the contact clamp strip. The right side of the top clamp has a section view cut-out, showing the copper wire on the contact clamp strip. With these clamps any soldering on the chip is avoided. **e** Electrical diagram for the multinuclear transmit and receive setup with three amplifiers. $^{1}H$ RF signals are passed through a high pass filter, $^{19}F$ and $^{13}C$ through their respective bandpass filters. The $^{1}H$ and $^{19}F$ signal paths are combined and connected to the probe. In the case of trinuclear experiments, a second combiner is introduced to feed the 13 C RF signals to the probe as well (blue dashed box). **f** Back view of the NMR chip holder. One of the three capillary connections is indicated in yellow. The fiber optic can be seen below the microfluidic ferrules (purple circle). The optical fiber is mounted in a slider, that can be moved up and down (purple arrow) to align the illumination spot with the NMR detection area.

combined in a single broad-band set up. Figure 1 shows the general set-up of an NMR magnet housing the home-built probe head that holds a microfluidic glass chip in the sweet spot of the magnet. The W-shape of the microfluidic chip allows for a dual-syringe operation, in which the external channels are inlets and the central one is outlet. The 3D-printed chip holder allows for the efficient integration of microfluidic, electronic and sample-illumination elements; the latter via a fiber-optics system transferring laser-diode light generated outside the magnet, non-invasively into the sample. Moreover, the flow set-up allows for rapid optimization of the various parameters influencing the photo-CIDNP efficiency, such as (relative) concentrations of the various components of target and photosensitizer, illumination power

and flow-speed. In addition, the planar broad-band microcoil set up allows for transmission of multiple RF frequencies to a single transceiver microcoil. Hence, complex heteronuclear and multidimensional NMR experiments can be executed. Uniquely, we now exploit two high-band channels in the single microcoil set-up, permitting the synchronous application of $^{1}H$ and $^{19}F$ Larmor frequencies within one pulse sequence. Here, we focus on a set of $^{19}F$ NMR spectroscopy experiments demonstrating the excellent mass, concentration sensitivity and multinuclear NMR applicability of this microcoil set up. We manage to perform even triple-nuclear $[^{1}H]^{19}F$-$^{13}C$ heteronuclear one-bond as well multiple-bond correlation experiments with a single radiofrequency (RF) - transceiver microcoil. Moreover, we performed 2D experiments

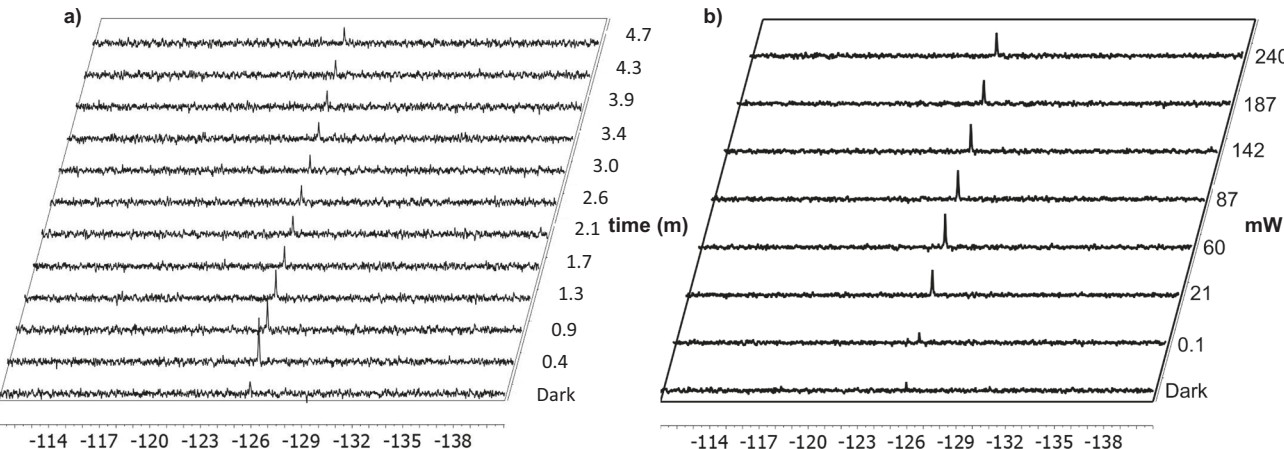

**Fig. 3 | Photodegradation of stopped-flow sample and optimization of the laser power. a** Collection of photo-CIDNP $^{19}$F-NMR spectra (from 0.4 to 4.7 minutes) of 100 mM of p-fluorophenol in the presence of 20 mM of FMN as photosensitizer under stopped-flow conditions. Spectrum 1 corresponds to the (control) experiment in the absence of light. The experiment time for each spectrum is 26 seconds (16 scans) and the vertical axis indicates total experiment time in minutes. **b** Collection of photo-CIDNP $^{19}$F-NMR spectra of 100 mM of p-fluorophenol in the

presence of 20 mM of FMN as photosensitizer under continuous flow conditions at a flow rate of 2 μL/min and with an increasing light intensity (from 100 mA to 700 mA). The output power (mW) out of the optical fiber is indicated in the vertical panel for the different light intensities. Only the absolute intensity spectra are shown; the commonly used difference spectra (light-dark) in photo-CIDNP experiments are not applicable here as in the dark practically no signal is observed.

including two high-band frequencies for 1D & 2D heteronuclear hyperpolarized $^{19}$F-$^{1}$H experiments in the lower mM concentrations range.

## Results

Below, our results are organized as follows: First, we describe in detail the microfluidic NMR setup as shown in Fig. 2, which consists of a microfluidic non-resonant NMR-chip system integrated into a tailor-made holder that facilitates electronic and fluidic connections as well as the optical fiber transmitting light from an external laser diode. Second, the performance of photo-CIDNP with planar spiral untuned microcoils is evaluated, including the on-flow optimization of experimental parameters such as light intensity and relative concentrations of components. Third, 1D and 2D mono-, di- and trinuclear $^{19}$F-NMR experiments on planar microcoils are described. Further, multinuclear 2D photo-CIDNP NMR experiments on continuous flow in the illuminated NMR chip with planar microcoils are presented, and the enhancement factors, and the performance are put into context of state-of-the-art NMR systems. Finally, a systematic investigation of the photo-CIDNP effect in a series of different sample concentrations reveals the non-linear relationship between target concentration (at fixed target/sensitizer ratio) and enhancement factor.

### System setup and holder design

Although microcoils are relatively robust in their handling, the importance of a proper and versatile holder can hardly be undervalued with some illustrative examples already published for Helmholtz[52], spiral[14,26,53], stripline[54], and microslot set ups[55]. The NMR chip holder for our planar spiral NMR-on-a-Chip is shown in Fig. 2 where the various connection points with the fluidics, electronics, and light source are laid out (See also Supplementary Figs. 1 and 2). Some key points of the NMR probe setup are provided in the caption of Fig. 2, while a detailed description of the holder itself is provided in the methods section. The planar spiral microcoil is shown as a copper disc. The chip features eight square contact patches for wire bonding and electrical connection. In this setup, only two patches are connected. The microfluidic chip has three fluidic connection ports. In the setup up shown here they are used as two inlets (solid yellow arrow) and one outlet (dashed yellow arrow) (Fig.1); in some experiments in this paper, the flow was

inversed with a single central inlet port and the outer ones functioning as outlet as mentioned above.

### Photo-CIDNP NMR-on-Chip and on-flow optimization of experimental conditions

We carried out photo-CIDNP investigations on p-fluorophenol, our model molecule since its $^{19}$F nuclide is highly polarized in the presence of flavin mononucleotide (FMN) according to our previous result and other reports in literature[44,56]. The starting concentration of p-fluorophenol was 100 mM with 2.5 nanomole (0.28 microgram) of material in the detection volume of the planar microcoil as required in our previous systems configuration[49,57–60]. A starting target molecule/photosensitizer ratio of 5 was used following our optimized conditions found for solenoidal microcoils. Accordingly, 100 mM of p-fluorophenol was irradiated in stopped-flow conditions in the presence of 20 mM of FMN (0.24 μg, 0.5 nmol). The dramatic enhancement of the NMR signal when compared to the non-illuminated spectrum (Fig. 3, left, the second and first spectrum of the array, respectively), is a clear illustration of the functioning of photo-CIDNP experiments on a planar microcoil set up. It confirms the applicability of the system to high optical density solutions, which are unreachable in photo-CIDNP experiments in conventional NMR tubes where the flavin concentration is typically restricted to the sub-mM range (ca. 0.2 mM)[61]. Next, we investigated the photodegradation of the FMN (Fig. 3, left, the whole array). The results shows that more than 50% signal is lost after ~100 seconds of uninterrupted light irradiation in our experimental conditions, illustrating the expediency of continuous flow regime, to avoid degradation and concomitant accumulation of photo-degraded flavin in the detection region and maintain a constant active photosensitizer concentration. The simplicity of the stopped/continuous flow setup allows to optimize experimental parameters quickly and iteratively for maximum photo-CIDNP, such as the light intensity. We incrementally increased the current controller output from 0 mA (dark) to 700 mA (which corresponds to 240 mW, Fig. 3, right), reaching highest signal at 300 mA (corresponding to 60 mW output power from the optical fiber and 8 mW on the NMR active volume (See Method section)).

We consecutively optimized the target-photosensitizer ratio exploiting the versatility of the W-shaped microfluidic chip with the two inlet channels for the in-situ mixing of solutions of target and

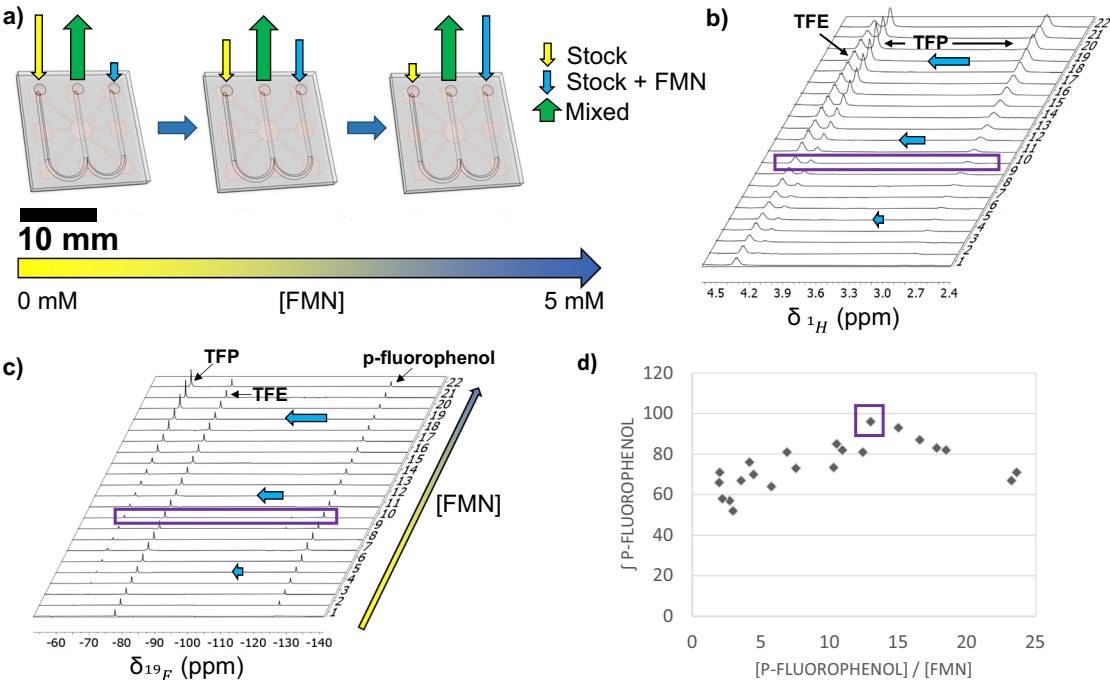

**Fig. 4 | Optimization of the p-fluorophenol-FMN concentration ratio. a** Scheme representing the backside of the NMR chip and the two inlets (yellow and blue arrows) from the two pumps placed outside the NMR magnet and one outlet (green arrow) (Figs. 1, 2). The blue arrow represents the inlet were FMN was present. The total flow rate was 2 μL/min (represented by the constant size of the green arrow), varying it from 0 to 2 for each syringe (represented by the different lengths for the yellow and blue arrows). **b** Collection of ¹H NMR spectra for different p-fluorophenol−FMN ratio when changing flow rates for the two pumps; the integral

values of both standards TFE and TFP were used to calculate the p-fluorophenol-FMN concentration ratio for **d. c** Photo-CIDNP ¹⁹F NMR spectra for the same conditions as in **b**. The purple rectangle indicates the spectra with a highest p-fluorophenol signal intensity. **d** Data points extracted from spectra shown in **c** for the integral value of p-fluorophenol as a function of the p-fluorophenol-FMN concentration ratio; The internal standard integral for TFE was taken as 100 units and the p-fluorophenol relative integral as a measure for photo-CIDNP efficiency.

photosensitizer. Two syringe pumps were placed outside of the NMR magnet. One of the solutions contained 300 mM of trifluoroethanol (TFE) (internal standard 1) and 10 mM of p-fluorophenol. The second solution contained 300 mM of TFE, 600 mM of trifluoropropanol (TFP, internal standard 2), 10 mM of p-fluorophenol and 5 mM of FMN as photosensitizer (see Supplementary Table 1). The p-fluorophenol−FMN ratio was then changed simply by changing the flow rate of the two syringe pumps from 2 μL/min to 0 μL/min for syringe 1 and synchronously from 0 μL/min to 2 μL/min for syringe 2, maintaining a constant flow overall rate of 2 μL/min, previously determined to be the optimal flow speed for mixing and time efficiency (Fig. 4a, Supplementary Table 1). Two different NMR spectra were collected for each flow rate combination, a ¹H NMR experiment (Fig. 4b) and consecutively, a photo-CIDNP ¹⁹F NMR experiment (Fig. 4c). The use of two internal standards, one of which (TFE) is present in both syringes in the same concentration and the second one is only present in one syringe (TFP), allows for the determination of the flavin concentration in each experiment, and therefore, the experimental p-fluorophenol−FMN ratio, directly from the ¹H NMR spectra (See Supplementary Table 1). The photo-CIDNP ¹⁹F NMR integral of p-fluorophenol for each p-fluorophenol−FMN ratio was then plotted to find the optimal ratio (Fig. 4d). Noteworthy, the integral values for the TFP peak tend to match with the expected values according to the selected flow rates of the two pumps, illustrating the efficient mixing of microchannels and good performance of the W-shaped chip and setup. The optimal p-fluorophenol−FMN ratio was 13:1 according to the maximum photo-CIDNP intensity for the p-fluorophenol (Fig. 4d). Notably, although an optimum sensitivity enhancement is observed from the titration experiment, the robustness of the photo-CIDNP experiment shown by significant enhancement (more than 50% of the maximum enhancement observed) over the wide range of ratios measured, is in

accordance with previous studies carried out in conventional photo-CIDNP experiments[62]. As we show later (vide infra), the optical density of the solutions is relatively high and also influences the perceived enhancement of the photo-CIDNP effect.

## 1D and 2D, ¹⁹F Multinuclear NMR on untuned spiral planar microcoils

Fluorine NMR[63], is becoming increasingly more and more recognized as useful nuclide to investigate, mainly due to its nuclear spin of ½, yielding sharp lines, a 100% natural abundance, a chemical shift dispersion in the order of more than 100 ppm and the high sensitivity due to a high magnetogyric ratio, close to that of ¹H. Last but not least, fluorine NMR is important because it is not an essential element in biological system, yet widely used in pharmaceutical compounds. Since ¹⁹F is considered a 'high-band' nuclide like ¹H, designated ¹⁹F hardware is necessary to perform experiments on this nuclide, particularly when performing simultaneous ¹H-decoupling and ¹⁹F acquisition; moreover, absence of PTFE (polytetrafluoroethylene) and other fluorinated materials for probe design is necessary. Such requirements hamper a more widespread use of ¹⁹F-NMR spectroscopy. The broadband aspects of a non-tuned circuit comprising a planar spiral microcoil enabled the detection of different nuclides in the full broadband range of Larmor frequencies (at 9.4 T from 61 to 400 MHz), with heteronuclear 2D experiments carried out on many combinations of nuclides both in coupled and decoupled mode[8]. We hypothesized before[8,25], and prove here that the broad-band nature previously reported by us provides the robustness to work with the planar microcoils also at higher magnetic field strengths and, moreover, with multiple high-frequency nuclides addressed synchronously.

The versatility and ease of use of the broad-band concept was tested in a 11.7 T NMR magnet as shown in Fig. 5a, c showing

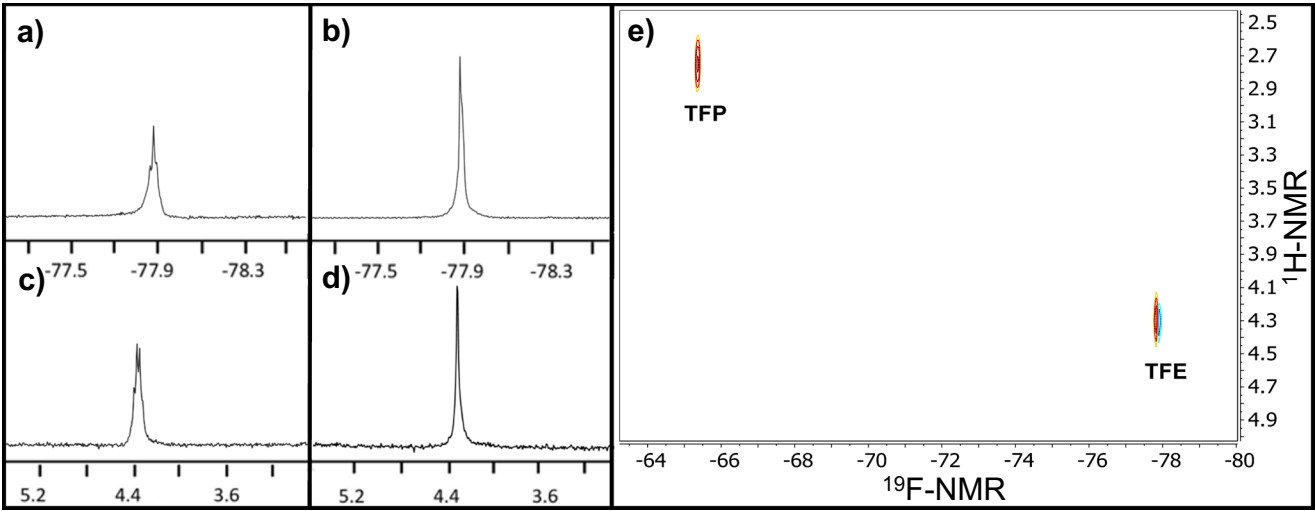

**Fig. 5 | 1D {1H}19F-NMR and {19F}1H-NMR and 2D 19F-1H-NMR with the untuned microcoil.** Left: (top) single-scan $^{19}$F-NMR spectra of neat TFE on the 25 nL spiral planar microcoil (342 nanomole) **a** proton-coupled and **b** proton decoupled, and (bottom) $^1$H-NMR spectra **c** fluorine coupled and **d** fluorine decoupled. **e** 2D $^{19}$F-$^1$H HMQC NMR on a mixture of TFE and TFP (1:1).

respectively spectra of 1D $^1$H NMR, (500 MHz Larmor frequency), and 1D $^{19}$F NMR (470 MHz Larmor frequency) performed on neat TFE (0.3 μmol). The two spectra were run consecutively without the necessity of any tuning to a certain frequency or any other setup modification. The spectra are well-resolved and the resolution obtained shows the $^3J_{H,F}$ coupling triplet pattern (8 Hz). Interestingly, the spectra can also be recorded in decoupled mode. Figure 5, left (b, d) shows single lines rather than multiplets, corresponding to the {$^1$H}$^{19}$F-NMR and {$^{19}$F}$^1$H-NMR spectra, respectively.

2D Heteronuclear NMR experiments are a valuable tool for molecular structure elucidation hence, we focused on testing the applicability of the broad-band spiral planar microcoil for high-band 2D $^{19}$F-$^1$H HMQC. A minor modification of the hardware (Fig. 2e) was required to combine the different radiofrequencies with two transmit channels and one transceiver channel on a single coil. Considering the small spectral separation between the $^1$H and $^{19}$F frequencies, two dedicated filters that cover a narrow frequency range were integrated in the hardware setup to avoid radiofrequency interferences (Fig. 2e), that possibly could damage the amplifiers. Firstly, the pulse width value was optimized in the indirect dimension following the *pwxcal* VnmrJ pulse sequence. Figure 5, right, depicts a heteronuclear 2D-experiment, $^{19}$F-$^1$H HMQC (heteronuclear multiple-quantum coherence), obtained in 24 minutes representing another proof of the potential of the broad-band set-up previously reported[8], but now also addressing two high-gamma nuclides at the same time.

**Multidimensional trinuclear $^{19}$F-$^{13}$C- $^1$H NMR experiments**
Recent developments in multidimensional NMR methodology are focused on increasing the spectral information content from single measurements, i.e., with multi-FID detection techniques that employ either a single receiver or multiple receivers[64], or with microcoil that can be integrated to multiple receivers[65]. Alternatively, we here demonstrate that the broad-band nature of a single spiral planar microcoil allows excitation of up to three different nuclides within a single 2D-experiment increasing further the frequency dimensionality, without the need for intricate and complicated pulse sequences or hardware. The experiments were carried out on neat TFE, corresponding to 0.3 μmole in the detection volume. However, as we measured only the natural-abundance (1.1%) $^{13}$C-labeled isotopes in a $^{19}$F$^{13}$C-HMQC or $^{19}$F$^{13}$C-HMBC experiment, in practice, the amount of detectable molecules in the detection volume was 3 nanomole. Fluorinated alkane-groups containing molecules that show

characteristic multinuclear *J*-coupling patterns as TFE[66], are ideal to prove the versatility of the broad-band NMR-chips[8]. Compared to previous experiments, an additional second combiner on the transceiver modules that mixes the two high-band $^1$H and $^{19}$F radiofrequencies with the $^{13}$C radiofrequency was included, to connect the transmitter and decoupler outputs from $^1$H and $^{19}$F to the $^{13}$C radiofrequency, and from there, to the single coaxial cable towards the probe head (Fig. 2e). The nutation curve for the calibration of the $^{19}$F 90° pulse width in these experimental conditions is shown in Supplementary Fig. S3a. The spectra shown below illustrate the effective performance of the relatively cheap and simple broad-band microcoil-on-a-chip in intricate and complex trinuclear NMR experiments.

In strong contrast to reported strategies with conventional probe-heads[64], the broadband nature of the RF circuit with the untuned planar spiral coil set up permits a straightforward execution of 2D- three-nuclide experiments. Supplementary Fig. 3b shows the $^{13}$C pulse width value optimization in the indirect dimension following the *pwxcal* VnmrJ pulse sequence. Hence, experiments such as {$^1$H} $^{19}$F-$^{13}$C HMBC (Fig. 6a, b, with and without $^1$H-decoupling during the acquisition, respectively) and {$^1$H}$^{19}$F$^{13}$C-HMQC (Fig. 6c, d, with and without $^1$H-decoupling during the acquisition, respectively) were run on a single planar spiral coil. Concerning the $^{19}$F-$^{13}$C HMQC experiments, it is important to recall that we observe only the fraction of the trifluoroethanol with the naturally abundant $^{13}$C isotope on the trifluoromethyl carbon, i.e., $^{13}$CF$_3$CH$_2$OH. The non-decoupled experiment (Fig. 6d, Supplementary Fig. 4) shows the two heteronuclear crosspeaks separated by a characteristically large $^1J_{CF}$ coupling constant of 290 Hz. In addition, each cross-peaks shows the fine structure of the $^3J_{HF}$ coupling of about 8 Hz. The $^1$H-decoupled {$^1$H}$^{19}$F-$^{13}$C HMQC experiment on the other hand (Fig. 6c Supplementary Fig. 5) shows still the two crosspeaks, but now with narrower peaks due to the removal of the methylene-protons' *J* coupling pattern. The 1D $^{19}$F spectrum has been enlarged and aligned on-top of the figure with the ppm scale of the inserts, to emphasize the non-symmetric positioning of the $^{13}$C isotope satellite peaks of the $^{13}$CF$_3$CH$_2$OH with respect to the most abundant, i.e., $^{12}$C-only, TFE peak in the 1D $^{19}$F spectrum. For the discussion of the $^{19}$F-$^{13}$C HMBC experiments it might be useful to recall that the spectra do not derive from the TFE described in the previous (HMQC) section but reflect the other $^{13}$C-containing isotope of trifluoroethanol, namely, the 1.1% fraction CF$_3$$^{13}$CH$_2$OH. The $^{19}$F-$^{13}$C HMBC experiment without $^1$H decoupling shows a characteristic skewed doublet-triplet pattern

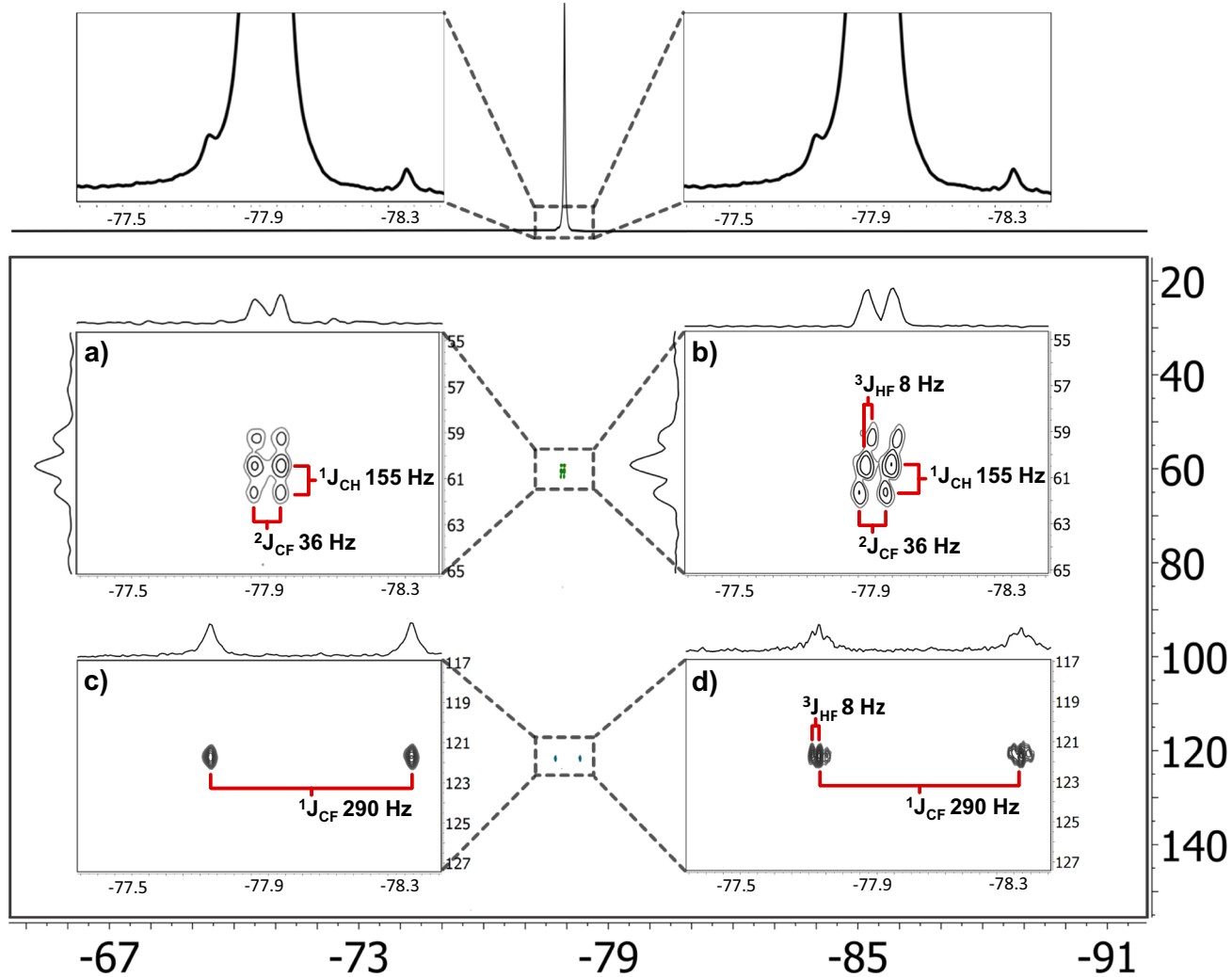

**Fig. 6 | Overlapped 19F-13C HMQC and 19F-13C HMBC on neat TFE (green crosspeaks in main spectrum).** The inserts show the $^1$H decoupled mode in **a**, **c** and without $^1$H decoupled mode in **c**, **d**. The inserts are aligned with the two blow-ups of the 1D spectrum on the main spectrum top, indicating the position of the $^{13}$C coupled and decoupled crosspeaks. **a** Shows {$^1$H}$^{19}$F-$^{13}$C HMBC, **b** shows $^{19}$F$^{13}$C-HMBC, **c** shows {$^1$H}$^{19}$F-$^{13}$C HMQC, and **d** shows $^{19}$F-$^{13}$C HMQC. The $^1J_{CF}$, $^1J_{CH}$, $^2J_{CF}$, $^3J_{HF}$ coupling constants values are shown, illustrating the good resolution of the spectra. $^{19}$F-$^{13}$C HMBC (**b**) shows a characteristic skewed doublet -triplet pattern

caused by $^1J_{CH}$, $^2J_{CF}$, $^3J_{HF}$ as previously reported[8]. The illustrative {$^1$H}$^{19}$F-$^{13}$C HMBC experiment (**a**) eliminates the $^3J_{HF}$ coupling pattern. Similarly, $^{19}$F-$^{13}$C HMQC (**d**) shows the $^3J_{HF}$ coupling pattern in both correlation peaks, and the related $^{19}$F-projection, whilst the {$^1$H}$^{19}$F -$^{13}$C HMQC (**c**) efficiently eliminates it. Note that we measured in neat TFE, so no lock is used, nor referencing to internal standards is done. The coupling constants are nevertheless field independent and can be determined from these spectra.

caused by the combined $^1J_{CH}$, $^2J_{CF}$ and $^3J_{HF}$ couplings as previously reported by us[8], demonstrating also the excellent spectral resolution (Fig. 6b, Supplementary Fig. 6). Moreover, the $^{19}$F projection on top of the insert shows the proton-coupling related fine-structures, i.e., the double triplet caused by $^1J_{CH}$ and $^3J_{HF}$. When the $^1$H decoupling is switched on, in the {$^1$H}$^{19}$F-$^{13}$C HMBC experiment, the skewed pattern disappears, and only two parallel lines are observed with only a doublet in the $^{19}$F-projection, reflecting the $^2J_{CF}$ coupling (Fig. 6a, Supplementary Fig. 7). This experiment indicates that a single microcoil is able to transmit RF pulses for three nuclides within a single pulse sequence, without having to tune to the individual Larmor frequencies. Note, here, the 2D insert crosspeaks are aligned under the central most intense $^{19}$F peak in the 1D spectrum on top, but the 2D inserts show formally the $^{12}CF_3$$^{13}CH_2OH$ signal in the projections. Since the all-$^{12}$C TFE is almost hundred times more abundant than $^{12}CF_3$$^{13}CH_2OH$, the fine structure of the second $^{13}$C isotope of TFE, together with its much smaller ($^{13}$C) isotope-effect shift, is lost in the relatively intense $^{12}CF_3$$^{12}CH_2OH$ peak in the 1D spectrum. For this reason, we chose to align the 2D inserts with the 1D

top spectrum, since the actual 1D $^{12}CF_3$$^{13}CH_2OH$ shift could not be determined under the large $^{12}CF_3$$^{12}CH_2OH$ peak.

## $^{19}$F-Photo-CIDNP planar spiral NMR mass sensitivity in 1D and 2D

In previous sections we have first introduced the microfluidic chip holder with photo-CIDNP option for improving the sensitivity of NMR, and second, the broadband performance of the planar spiral micro-coils for multinuclear multidimensional experiments. Here now, we do a thorough comparison of the mass sensitivity of the microcoil photo-CIDNP setup, including a comparison with a state-of-the-art 1200 MHz spectrometer equipped with cryoprobe, and, finally, we do a quantitative comparison of the heteronuclear 2D performance comparing the photo-CIDNP microcoil set up with a state-of-the-art iProbe at the same magnetic field strength.

First, to compare the mass sensitivity, the planar spiral coil set-up described above and operating in a 11.7 T magnet, was compared to data from a commercial 28.2 T spectrometer equipped with a 3 mm cryoprobe optimized for 200 μL sample tubes. On a 13 mM of p-fluorophenol and 300 mM TFE sample in water, a single-scan $^1$H-1D

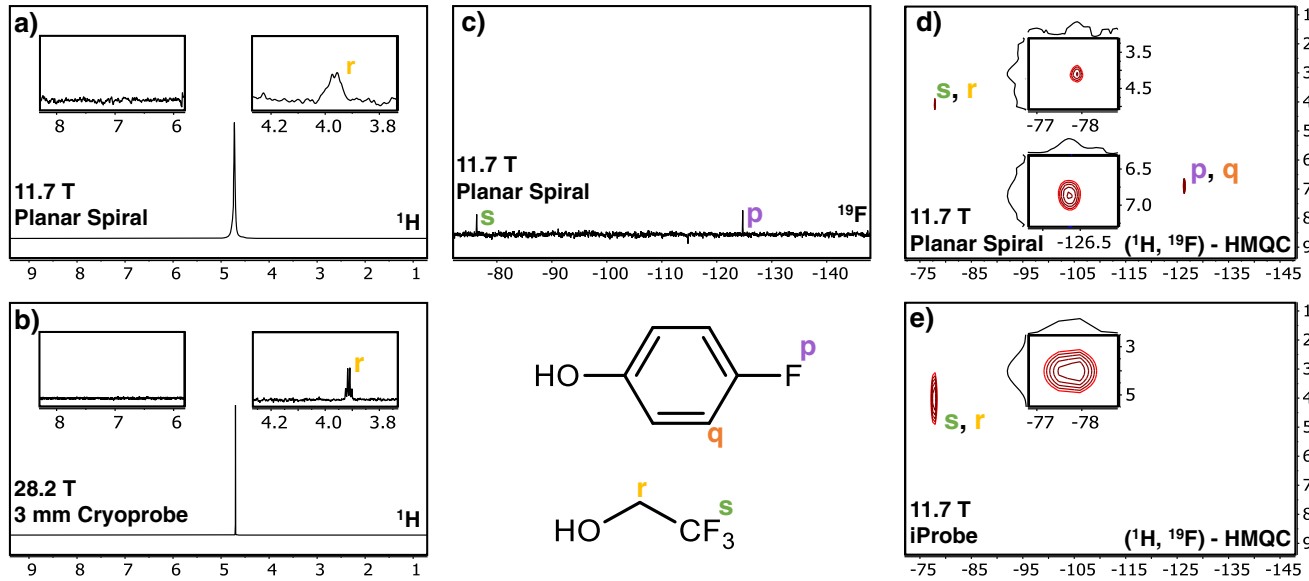

**Fig. 7 | Comparison of the photo-CIDNP broadband planar spiral microcoil setup with commercial 500 MHz probe and with a 1200 MHz NMR system.** **a** Single-scan 500 MHz [1]H spectrum of 325 picomole (13 mM) of p-fluorophenol, 7.5 nanomole (300 mM) of TFE and 28 picomole (1.1 mM) of FMN in $H_2O$ (nLOD$_{f,600}$ 1.0 nmol Hz$^{1/2}$), in the planar spiral microcoil. In the right inset the TFE quartet is visible (yellow **r**), but p-fluorophenol (left inset) is not observed. **b** 1200 MHz [1]H spectrum with the same amount of TFE and p-fluorophenol as in **a** dissolved in 200 μL $D_2O$ on a 3 mm cryoprobe. Also here, only the TFE quartet is observed after 1 scan (nLOD$_{f,600}$ 1.1 nmol Hz$^{1/2}$). **c** [19]F photo-CIDNP spectrum of 13 mM of p-fluorophenol, where the 300 mM TFE signal (green, **s**) is observed as well as the p-fluorophenol signal (purple, **p**) observed because of the hyperpolarization of the [19]F nucleus in 1 scan ((nLOD$_{f,600}$ 2.9 nmol Hz$^{1/2}$ for the thermally polarized TFE and

an nLOD$_{f,600}$ of 0.01 nmol Hz$^{1/2}$ for the hyperpolarized p-fluorophenol). **d** 2D photo-CIDNP [19]F HMQC spectrum at 2 μL/min of 13 mM of p-fluorophenol as in **c** using the untuned NMR chip. The [19]F-[1]H cross-peak of p-fluorophenol (**p**, **q**) as well as TFE's (**s**, **r**) are revealed in 32 scans. **e** Same amount of sample and experiment ([19]F-[1]H HMQC) as in **d**, but without photo-CIDNP, on a 500 MHz Bruker NMR system equipped with an iProbe. A similar NMR acquisition time does not render the corresponding p-fluorophenol peak, only the TFE can be detected. Note that all photo-CIDNP experiments were carried out in continuous light irradiation mode, simplifying the typical 2D photo-CIDNP pulse sequences[47], in combination with a continuous flow regime to avoid loss of signal because of the accumulation of photodegraded photosensitizer in the detection volume.

experiment with the microcoil was recorded at 500 MHz (Fig. 7a) showing a peak corresponding to an absolute amount of 7.5 nanomole of TFE. For comparison to the commercial 28.2 T NMR system, a 200 μL sample with 0.0015 mM p-fluorophenol and 0.038 mM TFE that corresponds to the same mass as measured in the microcoil, was measured (Fig. 7b), observing also a clear signal for the TFE. Both set-ups provide clear signals for the methylene protons of the TFE, with the higher resolution of the 1200 MHz spectrum evident from the H-F coupling showing a much narrower quartet peak than the 500 MHz spectrum. Importantly, in both spectra the p-fluorophenol signal is not observed. Following up, a single-scan experiment shows a clear peak of p-fluorophenol visible in the 470 MHz {[1]H}-[19]F Photo-CIDNP spectrum recorded on the non-resonant NMR chip (Fig. 7c); this corresponds to 325 picomole of p-fluorophenol. Note, the 28.2 T spectrometer used has no option to perform [19]F experiments with the 3 mm cryoprobe. This means that the introduction of photo-CIDNP on the untuned planar spiral coil operating at the relatively modest field of 11.7 T opens up the possibility of measuring mass-limited samples in the hundred(s) of picomole range. The mass sensitivity of the photo-CIDNP microcoil set up was next tested with the well-known antitumor drug 5-fluorouracil (5FU), which was irradiated in the presence of FMN. Two NMR spectra with similar signal-to-noise ratio (SNR) are shown in Supplementary Fig. 8, corresponding to a similar amount of fluorine, 5-fluorouracil (in the photo-CIDNP experiment, Supplementary Fig. 8a) and trifluoromethylphenol (in the broad-band circuit[8], Supplementary Fig. 8b). The results showed that the combination of broad-band circuits with photo-CIDNP enables the acquisition of a {[1]H}-[19]F photo-CIDNP NMR experiment with a time gain with respect to just a broad band configuration of two orders of magnitude. Less than 100 seconds were required to detect 80 picomole of 5-fluorouracil, whilst 25 picomole (75 picomole of "F") of trifluoromethylphenol could be detected

after 7 hours of NMR experiment (at 9.4 T) as previously reported by us[8].

Recently, Lepucki and coworkers published a method to systematically compare the performance of various NMR microdetectors using a normalized limit of detection at 600 MHz, nLOD$_{f,600}$[67]. We used the [1]H spectrum of Fig. 7a, to determine the nLOD for our untuned microcoil (see Method Section for details). We found an nLOD$_{f,600}$ of 1.0 nmol Hz$^{1/2}$ for the thermally polarized [1]H NMR experiment as determined from the $H_2O$ signal. We also determined the photo-CIDNP enhancement, on a [19]F spectrum of 1 mM of p-fluorophenol and also the TFE is detected (64 scans, Supplementary Fig. 9). Relative to the TFE integral (set to 300), p-fluorophenol showed an integral of 76.9 which indicates an enhancement factor of 230. Finally, in a single-scan experiment, the photo-CIDNP enhancement improves the untuned coil limit of detection, down to several hundred picomoles (Fig. 7c). We suspect that improvements in the RF hardware, such as the combiner interface as well as the non-resonant coil interfacing to the RF electronics may result in an even better limit of detection, but that is out of scope for the current work.

To further quantify the performance of a non-tuned coil in combination with photo-CIDNP, a 2D photo-CIDNP [19]F-[1]H HMQC experiment was set up on the NMR chip system and compared to the same experiment with the same absolute amounts of TFE and p-fluorophenol on a 11.7 T Bruker system equipped with a state-of-the-art 5 mm-tube iProbe. Hence, a 2D Photo-CIDNP [19]F-[1]H-HMQC experiment was run in the NMR chip on the same sample as in Fig. 7c, which corresponds to 13 mM of p-fluorophenol and 1.1 mM of FMN in the presence of 300 mM of TFE. The 2D spectrum was obtained within 20 minutes (32 scans per increment) and the crosspeaks of both the p-fluorophenol and the TFE are visible (Fig. 7d). As the experiment was run with flow speed of 2 μL/min, in 20' in total, 40 μL of 13 mM of

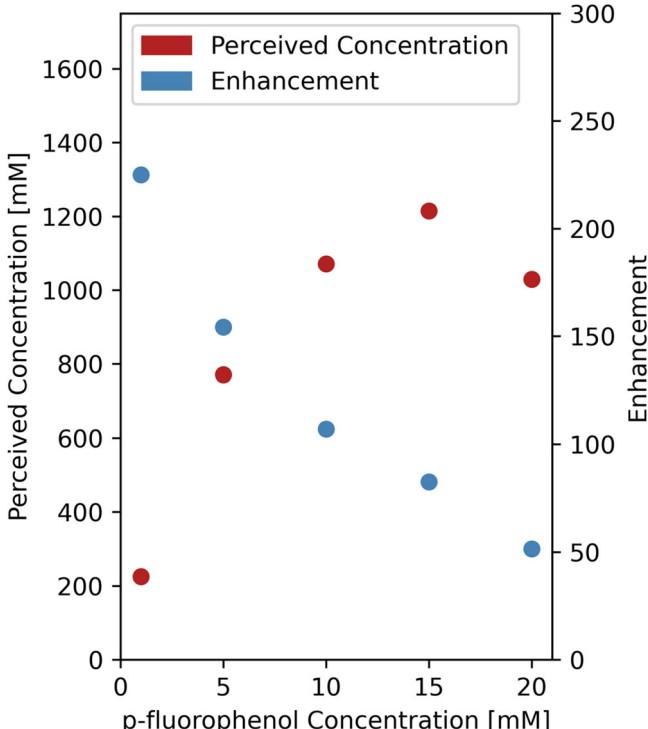

**Fig. 8 | The relation between the p-fluorophenol concentration and the perceived concentration as a result of photo-CIDNP enhancement.** Perceived concentration of the photo-CIDNP enhanced p-fluorophenol signal (red dots) as well as the enhancement factor (perceived concentration divided by the real concentration), showing a clear trend of higher enhancement at lower p-fluorophenol concentrations, and an optimum for maximum signal at 15 mM of p-fluorophenol. See supporting info for the raw data and normalization against the internal reference. Source data are provided as a Source Data file.

p-fluorophenol passed through the microcoil detection volume, corresponding to 0.5 μmole of p-fluorophenol. In contrast, on the commercial iProbe (Fig. 7e), in the 2D $^{19}$F-$^{1}$H HMQC the same amount, 0.5 μmole of *p*-fluorophenol dissolved in 0.5 mL (1 mM) remains undetected with the same experiment time than in the microcoil experiment.

In a final experiment we prepared a series of samples with the target molecule p-fluorophenol ranging from 1 to 20 mM and the photosensitizer at a constant ratio of 10% of the [p-fluorophenol]. We optimized the laser-diode current settings for each sample (see Supplementary Fig. 10–14) and then measured a photo-CIDNP enhanced spectrum. In all samples we used an internal standard of 300 mM TFE, which corresponds to a fluorine concentration of 900 mM. The integrals of the photo-CIDNP enhanced p-fluorophenol signals were then compared to the internal standard integral resulting in a so-called perceived [p-fluorophenol]. Dividing the perceived [p-fluorophenol] by the real [p-fluorophenol] in the sample then yields the enhancement factor. Interestingly, this enhancement factor is not a constant but shows a strong dependency in this series of samples. In Fig. 8 the perceived p-fluorophenol concentration and the enhancement factor are plotted against the target sample concentration. The perceived concentration in the series shows a clear increase going from 1 to 15 mM but then it drops. The enhancement factor on the other hand shows a clear decrease going from 230× to about 50×.

Typically, the dimensions in microfluidics are considered that small that light can penetrate readily through the sample. However, considering the high molar absorptivity of the photosensitizer FMN, the channel diameter in the 500 micron range and the flavin concentrations going from 0.1 to 2 mM, the effect can not be neglected.

Figure 8 shows that whilst the enhancement factor approximately linearly decreases in the higher concentration part of the series, the perceived concentration goes through an optimum. It is not straightforward to model and predict the enhancement factor to the experimental set up as multiple parameters are to be considered, as next to the Lambert-Beer parameters, for example also the geometry of the channel and the $B_1$ homogeneity will contribute to the perceived signal under varying conditions. As a matter a fact, the results presented in Fig. 4 are also related to the optical density aspects and the observed optimum target/flavin ratio is merely an empirical value for that specific sample conditions. We are currently looking into these aspects in more detail. In Supplementary Fig. 15 we also show a 0.5 mM sample of p-fluorophenol illustrating the relevant concentration limit for real samples.

## Discussion

Despite decades of groundbreaking developments, two major persistent factors still limit more widespread applications of NMR: the poor mass- and concentration sensitivity, as well as the intricate complex, and expensive hardware required for sophisticated NMR experiments. The development of in-house designed microcoils holds great potential for overcoming sensitivity, hardware, and costs limits. Pushing the mass limit of detection with microcoils is unfortunately often accompanied by a concomitant reduction in concentration sensitivity. Strategies to improve microcoils' sensitivity comprise various hyperpolarization techniques, that however reintroduce complexity in hardware and lead to high costs. Photo-CIDNP is one of the older and from a practical point of view, a more simply applicable hyperpolarization technique. Photo-CIDNP is particularly powerful for $^{19}$F NMR spectroscopy, as it is a very susceptible photo-CIDNP nucleus, leading to high nuclear polarization. Moreover, the $^{19}$F nuclide has a high magnetogyric ratio, close to that of $^{1}$H, a 100% natural abundance and a relatively wide chemical shift range. We here presented the use of a single untuned planar-spiral microcoil that can address different Larmor frequencies at the same time at a 11.7 T magnetic field strength (corresponding to 470 MHz $^{19}$F Larmor frequency) permitting also heteronuclear 1D & 2D experiments. A versatile W-shaped design of the microfluidics allows for a straightforward versatile use of a single-syringe or double-syringe set-up; invaluable for the intricate optimization of the multiple photo-CIDNP relevant parameters, such as illumination power, and the absolute as well as relative concentration of the photosensitizer and target molecules. We designed and 3D-printed a versatile chip holder to accommodate the electronic, fluidic and optical connections to the microfluidic NMR-chip. The flow set-up also alleviates one of the major drawbacks in photo-CIDNP experiments, which concerns the photodegradation of the photosensitizer and accumulation thereof in the detection volume. We address the two high-gamma nuclides, viz., $^{1}$H and $^{19}$F, and a third heteronucleus, i.e., $^{13}$C, allowing for complex trinuclear experiments such as {$^{1}$H}-$^{19}$F$^{13}$C multiple bonds correlation spectroscopy. 1D and 2D photo-CIDNP hyperpolarization schemes show excellent mass- and concentration sensitivity. Importantly, the concentration of components and the concomitant optical density is known to severely affect the distribution of light throughout the whole sample volume in conventional NMR tubes. With sample volumes in the order of submicroliters, this issue is alleviated, and even concentrated samples of highly absorbing compounds can readily be managed[44,59]. However, it can not be neglected as shown here. In fact, the trend of lower enhancement factor when measuring at increased concentrations is related to the optically density of the solution at the laser-diode wavelength. Similar to the more well-known inner-filter effect in fluorescence spectroscopy, increase of the concentration of the sensitizer in the case of photo-CIDNP experiments therefore does not by definition lead to a higher perceived signal. The combination of microcoils, with broadband set-ups in particular, and hyperpolarization is a promising step

forward in the development of NMR applications, ranging from small molecule analysis to drug screening that might find use in older as well as emerging hyperpolarization schemes[29], e.g., Xe-pumped[68], or nanodiamond NV centers NMR[69,70]. When spectral resolution and concentration sensitivity are the main requirement, the commercial NMR spectrometers are hard to beat, but for mass-limited samples and/or experimental set-ups where flow or small volumes are required, microcoil set ups have great advantages. Moreover, the use of microcoils in non-tuned RF circuits has the additional advantage that less common multi-nuclide combinations in advanced multidimensional experiments (i.e., {$^1$H}-$^{19}$F$^{13}$C HMBC/HMQC) can be investigated, which on standard setups, such as normal probes or the typically HCN-dedicated cryoprobes, often are not available.

## Methods

### Materials

All the samples (trifluoroethanol (TFE), trifluoropropanol (TFP), $^{13}$C-labeled acetic acid, Flavin mononucleotide (FMN), p-fluorophenol and 5-fluorouracile) were purchased from Sigma Aldrich and directly dissolved in 100% $H_2O$ to the concentrations specified for each experiment. $D_2O$ was only used for the experiments performed in the 28.2 T spectrometer and on the iProbe at the 11.7 T spectrometer.

### Experimental setup

**NMR probe base**. A decommissioned, non-functional 500 MHz standard bore Broad Band MAS probe was used as a platform. Considering the broad-band property of the planar coil in an untuned circuit[8], all the tuning and matching electronics are not necessary, and they were removed from the probe together with the pneumatics and Magic Angle Spin (MAS) mechanics. This leaves a probe frame with only the RF transmission lines and BNC connectors in place. The former $^1$H channel was used to connect to the NMR microcoil. The MAS probe skeleton provides an ideal base for attaching homebuilt microcoil hardware, but in principle any old probe body would suffice and the major point to keep in mind is the height of the 3D-printed holder to allow the chip to be well centered in the sweet spot of the magnet.

### Microfluidic planar spiral NMR chip

The experiments were performed using a glass microfluidic chip (Micronit) containing a planar spiral NMR microcoil similar to our previous design[8]. The NMR coil consists of 32 turns of copper of 20 μm width, 20 μm height and 20 μm spacing. The copper layer has 8 copper pads, of which 2 are used for coil connection. One patch is connected to the outermost coil turn, the innermost turn is connected via a wirebond to the opposite patch (red circles, Fig. 2). The other patches are unconnected but could be used to tap the coil windings. The microfluidics consist of a two inlet, single outlet channel (yellow circle, Fig. 2c). Using the double inlets, the photo-CIDNP dye/molecule ratio is easily optimized on-flow. Fused silica capillaries for continuous flow measurements are connected to the chip using a ferrule system. The middle outlet channel runs underneath the NMR coil. The coverslip thickness which separates the NMR coil from the sample is 100 μm thick. The detection volume under the coil was calculated based on the dimension of the channels and taking the inner diameter of the microcoil (250 μm), which is at a distance of 100 μm from the channel. MRI is most useful technique to verify microfluidic chip dimensions[57], and Supplementary Fig. 16 shows two cross sections of the microfluidic chip. Note: the exact number of detected spins also depends on the flow speed used in the experiments, as after the excitation pulse, and during the acquisition time, part of the excited volume will flow out of the detection volume.

### NMR chipholder

To use the microcoil NMR chip in a conventional standard bore NMR system, a custom chip holder was designed using a CAD program (SolidWorks 2019, Dassualt Systèmes). The parts were 3D-printed using a Formlabs Form 2 stereolithography printer. The chip can be seen mounted inside the holder, clamped in place by the Top Clamp, using a nylon M3 screw at each corner (Fig. 2d, Supplementary Fig. 1). Several iterations of the NMR chip holder led to the final design, for which the file is freely available (see data availability section). Initially, the holder used M2.5 screws, but the very fine thread caused problems in repeated use. Moreover, the acrylic 3D-printed material is brittle, which makes finely detailed post-printing modifications challenging. To increase the robustness of the setup, the final design uses M3 screws in all locations that can be repeatedly loosened and fastened, such as for the chip holder and the optical fiber slider. The M3 threads are also more easily tapped in the printed material. Larger threads may be even 3D-printed directly, but the narrow-bore (4 cm diameter within the shim coils) magnet puts physical limitations that do not readily allow larger hardware components to be used.

The narrow-bore also places constraints on fluidic and optical connections. The fluidic capillaries and optical fiber need to make a sharp bend to interface perpendicular to the NMR chip surface. In earlier designs we have experimented with side-entrance of the capillaries, but that concept suffers from easy breaking of the chips' glass top layer. Therefore, to relieve the strain on the connections, two off-center iterations of the NMR chip holder were tested. One had the sample center 2.5 mm offset and one with a 5 mm offset, radially from the magnet center. With the 5 mm off set holder it appeared very hard to shim, despite the fact the shim set is typically capable of shimming up to 10 mm diameter NMR tubes, so up to 5 mm out of the center. In practice, the 2.5 mm offset holder gave better NMR results and connector strain was sufficiently relieved. The resulting NMR chip holder and top clamp make replacing NMR chips easy and reproducible. In the final iteration, the optical fiber was mounted in a sliding mount, which also makes adjusting the illumination spot easy to be focused on the active NMR volume for optimal Photo-CIDNP conditions.

Electrical connection to the chip is made by two 0.5 mm enameled copper wires that are soldered to the coaxial transmission line and the probe frame (ground), respectively. The wires connect to two copper contact strips in the Top Clamp, one on each side. The wires are pushed onto the copper contact strip by the outer M3 screws. The copper strips have an indentation at the inward facing end that is pushed onto the NMR chip contact patch, using the inner M3 screws ((red circle, Fig. 2d, red box, Fig. 2b). The Top Clamp and screw clamping system make swapping NMR chips easy. The microfluidic connections to the chip are made via fused silica capillaries, that are pushed on the dual inlet and single outlet using a ferrule system (yellow box, Fig. 2b, yellow circle, Fig. 2f). For Photo-CIDNP experiments, the sample is illuminated from the back by an optical fiber (purple box, Fig. 2b, purple circle, Fig. 2c). The optical fiber is mounted in the Fiber Holder Slider (purple circle, Fig. 2f), which can be moved along the outlet channel axis, to align the illuminated area with the NMR detection area (purple arrow, Fig. 2f)

### Multinuclear broadband setup

The broadband RF behavior of the planar spiral microcoils in a capacitor-free non-tuned circuit were used for multinuclear NMR experiments on $^1$H, $^{19}$F and $^{13}$C nuclei. The two high-band channels ($^1$H and $^{19}$F) were fed into a combiner that is connected to the probe (Fig. 2e). In the case of trinuclear experiments, a second combiner was introduced to feed the $^{13}$C signal to the probe as well. Filters were placed in front of the amplifier channels, high pass for $^1$H and bandpass for $^{19}$F and $^{13}$C (Fig. 2e) (500-18, 470-18 and 135-35 respectively).

### Photo-CIDNP experiments

The light source used for the experiments consisted of a 450 nm laser diode with a 1.6 W maximum output power. The optical power is decreased to the milliWatt range by using a controller that regulates

the current through the diode. The current controller (Thorlabs, LDC240C) can be switched on and off to regulate the duration and intensity of the light manually. A temperature controller (Thorlabs, TED200C) is also used to ensure no variation of the output light power. A photometer PM100D (Thorlabs) was used to monitor the desired power from the light beam coming out of a 400 μm (internal diameter) optical fiber. An optical fiber (Thorlabs, FT400EMT) of 400 μm diameter is used to guide the light from the laser diode to the sample. All photo-CIDNP experiments were carried out in continuous light irradiation mode, in combination with continuous flow regime (except for the detection of 5-fluorouracil which was acquired in stopped-flow conditions). The enhancement factor experiments for Fig. 8 were acquired with a 600 μm (internal diameter, FT600EMT) optical fiber and optimal output powers were determined to obtain maximum signal (Supplementary Figs. 10–14).

## NMR experiments

The experiments on planar spiral microcoils were carried out in an 11.7 T narrow-bore Oxford instruments magnet (500 MHz $^1$H Larmor frequency and 470 MHz $^{19}$F Larmor frequency) INOVA NMR spectrometer (Agilent). All experiments were acquired in non-locked mode, adjusting shims on the $^1$H solvent peak. Standard pulse sequences from the VNMRJ software were used for all NMR experiments (Version 2.1 Revision A). Spectra were processed with VnmrJ (Supplementary Figs. 10–14) or MestReNova 14.2.2 software (Mestrelab Research, S. L., Spain).

Control experiments were acquired on commercial NMR probes taken as a reference for the mass-sensitivity analysis. An NMR tube with the absolute amount detected on the spiral planar microcoil and the corresponding volume of deuterated solvent was prepared for every experiment. For 1D $^1$H NMR experiments, a 1200 MHz Bruker NMR system equipped with a 3 mm Cryoprobe was used, whilst for 2D NMR experiments, it was a 500 MHz Bruker Advance Neo equipped with a 5 mm iProbe. For 1D $^1$H NMR experiments, 325 picomole of p-fluorophenol and 7.5 nanomole of TFE were dissolved in 200 μl of $D_2O$, resulting in a concentration of 1.6 μM of p-fluorophenol and 37.5 μM of TFE. For the 2D Photo-CIDNP {1H}$^{19}$F-$^1$H HMQC, 0.52 μmol of p-fluorophenol and 12 μmol of TFE were dissolved in 500 μl of $D_2O$, resulting in a concentration of 1 mM and 24 mM, respectively.

The MRI images were obtained on a 14.1 T 600 MHz Bruker AVANCE III spectrometer equipped with triple 60 A GREAT amplifiers and a micro5-microImaging probe body with a 1 cm 1H coil insert. One microfluidic NMR chip was sacrificed and cut such that the central microfluidic part fitted in the sensitive part of the imaging coil. MSME experiments were run to visualize two orthogonal cross-sections of the microfluidic channels.

The standard VnmrJ PWXCAL pulse sequence which is normally used to calibrate the pulse width characteristics of the probe's decoupler channel in indirect detection or triple resonance experiments was employed for acquisition parameters optimization. Neat TFE was used for the optimization of the proton indirect detection parameters for the $^{19}$F$^1$H-HMQC NMR experiments, whilst neat acetic acid-1-$^{13}$C was used for the indirect calibration of the 90° pulse width of the $^{13}$C channel for the $^{19}$F-$^{13}$C HMQC, for the $^{19}$F-$^{13}$C HMBC NMR experiments and for the trinuclear experiments. For the $^{19}$F-$^1$H HMQC experiment the standard delays used were calculated from the three-bond ($^3J_{FH}$ = 8 Hz) proton-fluorine coupling constant. For the heteronuclear $^{19}$F$^{13}$C experiments the corresponding delays were adjusted for the larger one-bond ($^1J_{CF}$ = 290 Hz) and two-bond ($^2J_{CF}$ = 36 Hz) carbon–fluorine coupling constants.

1D decoupled {1H}$^{19}$F-NMR and {$^{19}$F}$^1$H-NMR experiments were run on neat TFE (0.34 micromole) in stopped-flow conditions with a single-scan. 2D $^{19}$F-$^1$H NMR was performed on a mixture of TFP and TFE (1:1) in stopped-flow. The acquisition parameters are summarized in Supplementary Table 2. Multidimensional trinuclear $^{19}$F-$^{13}$C-$^1$H NMR

experiments, ({1H})$^{19}$F-$^{13}$C HMQC and ({1H})$^{19}$F-$^{13}$C HMBC, were run on neat trifluoroethanol (342 nmol) in stopped-flow regime. The acquisition parameters are summarized in the Supplementary Table 2.

1D $^{19}$F photo-CIDNP NMR experiments. Proof-of-concept $^{19}$F photo-CIDNP on spiral microcoils. For experiments shown in the first section for proving the concept of $^{19}$F photo-CIDNP on spiral planar coils and for the optimization of experimental parameters, 100 mM of p-fluorophenol and 20 mM of flavin mononucleotide were dissolved in 100% $H_2O$ and pumped in continuous flow regime at 2 μL/min. The number of scans used was 16. When the light current was varied from 0 mA to 700 mW, 300 mA was found to be the optimum value under these conditions, which corresponds to 66 mW output power at the fiber end. The output power (mW) for every light current was measured with a photometer PM100D (see above). The fiber was positioned 3.5 mm from the chip. The internal distance of the chip surface to the channel center is 0.76 mm. Because the light from the 0.39 NA fiber spreads out in a cone, roughly 13% of light falls onto the NMR active volume. This would correspond to 8 mW of optical power on the NMR volume. This light output value was the same employed for all experiments of this section.

Titration NMR experiments on flow with two inlets/syringes/pumps. Optimization of target-photosensitizer ratio. To obtain useful parameter information for concentration-sensitive experiments, lower concentrations of substrate were tested for the target and photo-sensitizer in comparison to the proof-of-concept experiment (vide supra). For the experiments carried out to optimize the target-photosensitizer ratio, two syringes placed in two different pumps were filled in with the following: Pump 1: A solution of 10 mM of p-fluorophenol and 300 mM of trifluoroethanol. Pump 2: A solution of 10 mM of p-fluorophenol, 5 mM of FMN as photosensitizer, 300 mM of trifluoroethanol (TFE), and 600 mM of trifluoropropanol (TFP). TFE and TFP were used as internal standards to calculate the real flow rate values of the setup for the different experimental conditions. The flow rate values for both, pump 1 and pump 2 were varied from 0 to 2, acquiring NMR spectra for the ratio values shown in Supplementary Table 1. $^1$H NMR spectra and (light) photo-CIDNP $^{19}$F NMR spectra were acquired for every flow rate combination. Integrals value for calculating the real TFP concentration (and therewith the real FMN concentration and thus the p-fluorophenol/FMN ratio) were measured in the $^1$H NMR spectra. The data points from Fig. 4d were obtained in an iterative mode, changing the flow rates for both pumps according to Supplementary Table 1, waiting ~5–10 minutes for reaching the expected concentration ratios for the two standards according to the observed integrals, and starting the acquisition of $^1$H NMR and photo-CIDNP $^{19}$F NMR spectra, resulting an efficient method for optimizing experimental parameters. Noteworthy, 1.1 mL of sample was used for the whole titration experiment, corresponding to only 11.5 μmol of p-fluorophenol. The number of scans used for the $^1$H NMR experiments was 128, whilst for the $^{19}$F NMR experiments was 64.

1D $^{19}$F photo-CIDNP NMR experiments. Mass-sensitivity and signal enhancement. For the experiments carried out for the mass-sensitivity comparison, a solution of 13 mM of p-fluorophenol in the presence of 1.1 mM of FMN and 300 mM of internal standard, TFE, was irradiated in continuous flow at 300 mA (66 mW, 8 mW on the active sample volume). The 1D $^1$H NMR spectrum shown in Fig. 7a was acquired in the same experimental conditions. The number of scans was 1, corresponding to the detection of 325 picomole of p-fluorophenol. The same sample was used for the nLOD calculation[67]: A single-scan $^1$H 500 MHz spectrum was used, with a bandwidth of 8013 Hz, SNR (MNova) of 6251 on the $H_2O$ peak (55.55 M, 2 protons per molecule) 8000 spectral points. For the 1200 MHz, the $^1$H signal of TFE was used, taking into account the 1:3:3:1 splitting pattern. The spectrum was recorded with a bandwidth of 14705 Hz, 65536 points, and an SNR of 25.0. In Fig. 7c the bandwith was 89259 Hz and the number of points was 131072. The SNR on the TFE triplet signal was 7.1 and 10.8 on the

p-fluorophenol triplet of triplets. All data were scaled to the 600 MHz standard suggested by Lepucki et al. corrected for the number of nuclei per molecule and multiplet intensities. For the signal enhancement, we used spectrum in which both TFE and p-fluorophenol are resolved (Supplementary Fig. 9). To determine the enhancement factor, the TFE peak integral was set to 300 and the resulting p-fluorophenol integral was 76.9. With a TFE:p-fluorophenol fluorine atom ratio of 69, this leads to an enhancement of 231.

2D $^{19}$F photo-CIDNP HMQC NMR experiments. A continuous flow solution (2 µl/min) of 13 mM of p-fluorophenol in the presence of 1.1 mM of FMN and 300 mM of TFE was continuously irradiated with a light current of 300 mA (66 mW). The acquisition parameters are summarized in Supplementary Table 2.

Enhancement factor and perceived concentration determination experiments. A fixed FMN:p-fluorophenol ratio of 1:10 was used. Different dilutions from a stock solution were made, containing 20, 15, 10, 5, and 1 mM of p-fluorophenol, respectively, 2, 1.5, 1, 0.5, and 0.1 mM FMN. Samples were measured on-flow, with a flowrate of 4 µL/min. Optimal laser intensity was determined on-flow and the optimum point was used for Fig. 8. Individual plots of the laser optimization are shown in Supplementary Figs. 10–14.

## Data availability
All relevant data generated in this study have been deposited in the ZENODO database under the accession code 7930396[71]. Source data are provided with this paper.

## Code availability
The code for Fig. 8 in this study has been deposited in the ZENODO database under the accession code 7930396[71].

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

## Acknowledgements

M.V.G. thanks the Spanish Government (AEI/FEDER UE) for funding: Project PID2020-119636GB-I00, funded by MCIN/AEI/10.13039/501100011033 and CTQ2017-84825-R, and Junta de Comunidades de Castilla-La Mancha (SBPLY/21/180501/000114). S.B. and A.H.V. acknowledge financial support from QuantERA (NanoSpin, project nr. 13N14809). This project was supported by uNMR-NL, the National Roadmap Large-Scale NMR Facility of the Netherlands (NWO grant 184.032.207). Volker Lehmann (Bruker) is acknowledged for help with the MRI experiments.

## Author contributions

M.V.G. initiated and designed the work, carried out all NMR experiments, analyzed and interpreted the data, and wrote the manuscript. S.B. designed and fabricated the probe and holder, executed NMR experiments, analyzed and interpreted the data, and wrote the manuscript. A.H.V. initiated and designed the work, carried out NMR experiments, analyzed and interpreted the data, and wrote the manuscript.

## Competing interests

The authors declare no competing interests.
