## [Peer review file · Nature Communications]

REVIEWER COMMENTS

Reviewer #1 (Remarks to the Author):

This is a very impressive paper from leaders in the field. The science is of very high quality. I am particularly impressed they performed the miniaturization in a broad band fashion. The comparison to 1.2 GHz is also impressive.

I recommend acceptance after a couple of minor considerations.

It is clear that the CI-DNP improves the signal. Is it possible to state the detection limits in the abstract. If I understand it correctly you already have nLOD_{f,600} of 1.48 nmol Hz^{1/2} which is incredibly impressive for a single sided broadband coil. Then you beat this by 450 times with the CI-DNP fluorine. Therefore would it be worth mentioning either "broadband coil limit of detection, down to several hundred picomoles" or the nLOD_{f,600} for ¹⁹F with CI-DNP in the abstract ?

Is it possible to please state the calculations as to how you get 25nL for the detection volume, i.e. the channel diameter across the coil surface and approximate distance from the channel to coil surface, and what region of the coil you use for the active region (if the calculation assumes only the coil I.D. of 250um, can you please note this in the paper). The nutation curves are very impressive so it is clear the channel has been optimized in the sweet spot, but exact dimensions are important to make it easier for others to follow, replicate and compare to similar and other coil designs.

Can you please cross check that ¹⁹F¹H-"HMQC" is the correct term. As the ¹H and ¹⁹F are not directly bonded are one of the terms COSY, TOCSY or HOESY more appropriate vs HMQC. I am not an expert at this so I may be incorrect. If HMQC is the correct term please ignore this comment.

Reviewer #2 (Remarks to the Author):

Gomez et al present a study of hyperpolarized NMR using the chemically-induced dynamic nuclear polarisation (CIDNP) effect in combination with a planar micro coil NMR detector and a microfluidic flow system. They show that their setup is capable of running advanced multinuclear NMR methods requiring simultaneous irradiation at the ¹H, ¹³C, and ¹⁹F nuclear Larmor frequencies at high magnetic field (11.7 Tesla). The combination of micro-NMR detection and CIDNP has been reported by some of the same authors before (Mompean et al, Nat Comms 2018, DOI: 10.1038/s41467-017-02575-0).

While the paper contains some interesting new results, it represents a largely incremental step over previous work. Moreover, the concept of "broadband" microcoils as presented in the manuscript is poorly motivated and lacks substance, as discussed in more detail below. For these reasons, I recommend rejection of the manuscript in its present form. Specifically, I have the following concerns and comments:

1. Compared to previously published work, the present manuscript introduces planar micro coils and a changed fluidic geometry. The authors claim that these planar coils have some special "broadband nature", due to which one can forgo the usual (complex) hardware needed for multiple-frequency tuning and matching. This assertion is repeated without evidence throughout the paper, implied by referring to "the broadband coil" etc. This does not, unfortunately, make it any more convincing. Microcoils do not defy the laws of

electrodynamics just because they are small.

2. The microcoil terminates a 50 Ohm transmission line with an impedance that is likely to be dominated by an inductance, which by experience lies in the range of a few nH for a coil of this size. Together with a small resistive component (say, 0.1 Ohm), this translates to an impedance at 500 MHz of about $Z=(0.1+5j)\text{Ohm}$, extremely far from a matched condition for any of the frequencies in question. As a result, the entire setup, including the 50 Ohm transmission lines, filters, and connecting cables, will exhibit a very large standing wave ratio, leading to a large fraction of the signal power being dissipated in the connection network. At the same time, the signal from the coil does not benefit from resonant amplification. Under these conditions, the preamplifier noise (rather than the Johnson noise from the coil) dominates, resulting in a loss in sensitivity. This is clearly visible in the results presented in Fig 3: from the ^1H spectrum (Fig 3d) one can discern a signal/noise ratio of about 30. Together with the proton concentration ($2 \times 10^{25} \text{M}$) contributing to this signal, and a line width of about 10Hz, one obtains a number limit of detection (nLOD) of about 20 nMol $\sqrt{\text{s}}$. The authors present a more favourable LOD calculation from the results in Fig 5a; however, it ignores the fact that the methylene multiplet is not resolved in the spectrum, and therefore is probably too optimistic. Detectors of similar volume have been presented in the literature with nLODs around 0.1 nMol $\sqrt{\text{s}}$ (e.g., van Bentum et al, J. Magn. Reson. 189, 104-113 (2007)).

3. The argument for the "broadband" nature of these coils needs to be substantiated, and the case for their use balanced against the tradeoff in sensitivity.

4. That said, the results presented by the authors in this and previous publications do show that in spite of this, NMR signals can be acquired successfully. There is merit in the simplicity of the approach, which contrasts with the complexity of coil tuning and matching, particularly if multiple frequencies are required. However, the implication that there is some magic property of these micro coils that makes them "broadband" is misleading at best if the severe tradeoff in sensitivity is not discussed alongside.

5. The present manuscript shows that multinuclear HMBC and HMQC experiments are feasible in this setup at thermal polarisation (without CIDNP) with highly concentrated samples (neat trifluoroethanol). While this is interesting as a demonstration of principle, it is unclear what the use case would be if such highly concentrated samples are necessary.

6. The discussion of sensitivity of the CIDNP results (Fig 5) should be done in terms of clear indications of the nLOD figures for each case. The comparison of signal amplitudes obtained by averaging different numbers of transients and under different conditions (field strength, sample volume, coil size, with/without CIDNP amplification) makes it impossible to discern the true advantages of the approach.

7. (Related point) The results section claims "... photo-CIDNP on the broadband planar spiral coil ... opens up the possibility of measuring ... samples in the hundred(s) of picomole samples". This is misleading, as this level of sensitivity has been demonstrated by multiple authors at the same detector size WITHOUT hyperpolarization.

Minor points:

8. The manuscript pages should be numbered.

9. Fig 1: instead of numbering the traces, indicate the time (left panel) or the diode current (right panel). The scale indicator "Light intensity" with units of mA does not make sense, and is confusing given that the numbers next to it refer to the trace number, not the diode

current.

10. Scheme 2 and Figure 2a would benefit from a size scale.

11. Fig. 4: axis should be clearly labelled, e.g., "¹³C Chemical shift [ppm]" etc

12. Fig 5: see point 11

Reviewer #3 (Remarks to the Author):

The manuscript 'Multinuclear 1D & 2D NMR with ¹⁹F-Photo-CIDNP hyperpolarization in a microfluidic chip with broadband microcoil' by Gomez and co-workers describes the integration of a new home-made device for sample illumination with the nanoliter-scale microfluidic broad-band NMR chip previously introduced, as well as the on-flow photo-CIDNP experiments performed to assess this device. This work is a continuation of previous developments of this team, dealing with broadband NMR micro-detection (ref. 8), microfluidics and photo-CIDNP (ref. 44), its aim being to combine microfluidic and photo-CIDNP approaches with broadband (trinuclear) NMR detection.

The device is ingenious, well designed and reveals good performances in terms of hyperpolarization by photo-CIDNP (applicability of the system to high optical density solutions) and of ¹H/¹⁹F(-¹³C) NMR. In the former experiments the practicality of the device is used to optimize the target-photosensitizer ratio. In the latter, heteronuclear inverse experiments show the capability of the microcoil to transmit ¹³C, ¹⁹F and ¹H rf pulses within a pulse sequence without individual tuning (true for HMQC/HMBC, but how to perform a HSQC?).

The bibliography appears to be complete, with few or no missing references, and there is enough details in the Methods part for the work to be reproduced.

Even if according to me the manuscript does not contain major scientific or technological breakthrough, the full integration and performances of the device make that this deserves to be published in Nature Communications.

Minor remarks:

- Beginning of the Results & Discussion part: in the sentence 'the performance of photo-CIDNP in spiral planar broad-band planar microcoils...' maybe remove one of the two 'planar'
- Figure 1 is not very clear: 1) the intensity scale of the peaks should be increased (even if it means that the spectra overlap, which is not a problem since they are offset), 2) for the sake of clarity on the right-hand series the light intensity should directly constitute the y axis, 3) the last sentence in the caption should be removed and made part of the main text only, as it is confusing.
- Caption of Fig. S2: please indicate the units (s, W, etc.)

REVIEWER

COMMENTS

Reviewer #1 (Remarks to the Author):

This is a very impressive paper from leaders in the field. The science is of very high quality. I am particularly impressed they performed the miniaturization in a broad band fashion. The comparison to 1.2 GHz is also impressive.

I recommend acceptance after a couple of minor considerations.

We thank the reviewer for his very positive comment and feedback. In particular, we acknowledge the reviewer values the direct comparison with the state-of-the-art 1200 MHz system. This is, in fact, in our opinion most illustrative and perhaps even more than the cumbersome and less hands-on definition of $nLOD_{f,600}$. In particular for the broader readership such a comparison is expected to be of more impact.

It is clear that the CI-DNP improves the signal. Is it possible to state the detection limits in the abstract. If I understand it correctly you already have $nLOD_{f,600}$ of 1.48 nmol Hz^{1/2} which is incredibly impressive for a single sided broadband coil. Then you beat this by 450 times with the CI-DNP fluorine. Therefore would it be worth mentioning either “broadband coil limit of detection, down to several hundred picomoles” or the $nLOD_{f,600}$ for ¹⁹F with CI-DNP in the abstract ?

We have followed the referee’s suggestion and added the $nLOD_{f,600}$ for ¹⁹F with photoCIDNP in the abstract. We have included this on our new data from the 13 mM experiment in figure 5. It is 0.012 nmol sqrt(Hz) for fluorine with photo-CIDNP. Regarding the enhancement factor, whilst revisiting the old data sets, we noticed that the 450 factor was overestimated because of a titration error. We apologize for that, and therefore repeated the experiments and made sure to add a double control for the quantification, by using 1D ¹H spectra in which the relative integrals of the water and TFE peaks corroborate an internal standard of 300 mM TFE in the new set of experiments. Most interestingly, we then noted a strong dependency of the enhancement factor as a function of the sample concentration, which we believe is most informative, and ranges up to 230 fold enhancement at 1 mM. Importantly, to obtain the highest signal from a sample, it is not the highest enhancement factor one needs to search for, but the highest apparent concentration. We have added this as a new graph to the main ms.

New figure 6. Perceived concentration of the photo-CIDNP enhanced parafluorophenol signal (red dots) as well as the enhancement factor (perceived concentration divided by the real concentration), showing a clear trend of higher enhancement at lower (FMN) concentrations, and an optimum for maximum signal at 15 mM pF. See supporting info for the raw data and normalization against the internal reference.

Is it possible to please state the calculations as to how you get 25nL for the detection volume, i.e. the channel diameter across the coil surface and approximate distance from the channel to coil surface, and what region of the coil you use for the active region (if the calculation assumes only the coil I.D. of 250um, can you please note this in the paper). The nutation curves are very impressive so it is clear the channel has been optimized in the sweet spot, but exact dimensions are important to make it easier for others to follow, replicate and compare to similar and other coil designs.

For determining the exact dimensions and volume we have sacrificed one chip cutting it such that the fluidic parts fitted into the coil region of micro5 probe (see photo below). We then recorded high-resolution MRI images of the microfluidic chip which have been added in the supporting information and we have added the following text to the methods section of the main text:

The detection volume under the coil was calculated based on the dimension of the channels and taking the inner diameter of the microcoil (250 um), which is at a distance of 100 um from the channel. MRI is most useful technique to verify microfluidic chip dimensions,[71] and supporting info figure S. shows two cross sections of the microfluidic chip. Note: the exact number of detected spins also depends on the flow speed used in the experiments, as after the excitation pulse, and during the acquisition time, part of the excited volume will flow out of the detection volume.

Figure S. micro-MRI orthogonal slices of a MSME experiments on a sacrificed microfluidic NMR chip filled with water, revealing the geometry and dimensions of the channels.

Can you please cross check that $^{19}\text{F}^1\text{H}$ -“HMQC” is the correct term. As the ^1H and ^{19}F are not directly bonded are one of the terms COSY, TOCSY or HOESY more appropriate vs HMQC. I am not an expert at this so I may be in correct. If HMQC is the correct term please ignore this comment.

We understand the confusion as typically the HMQC is used for heteronuclear 1-bond (e.g. $^1\text{H}^{13}\text{C}$ or $^1\text{H}^{15}\text{N}$) detection, however, we adapted the mixing timing such to allow for the 3-bond connection with the relatively small scalar coupling of $^3J_{\text{HF}}$ of 8 Hz to be observed. For clarity purposes we have added this explicitly to the methods section of the paper:

A mixture of trifluoropropanol and trifluoroethanol (1:1) in stopped-flow was used to run the $^{19}\text{F}^1\text{H}$ -HMQC experiment. The mixing time was set to 125 ms in order to select for the long range coupling constant of 8 Hz of the $^3J_{\text{HF}}$. Other acquisition parameters are in the SI Table 2.

Reviewer #2 (Remarks to the Author):

Gomez et al present a study of hyperpolarized NMR using the chemically-induced dynamic nuclear polarisation (CIDNP) effect in combination with a planar micro coil NMR detector and a microfluidic flow system. They show that their setup is capable of running advanced multinuclear NMR methods requiring simultaneous irradiation at the ^1H , ^{13}C , and ^{19}F nuclear Larmor frequencies at high magnetic field (11.7 Tesla). The combination of micro-NMR detection and CIDNP has been reported by some of the same authors before (Mompean et al, Nat Comms 2018, DOI: 10.1038/s41467-017-02575-0).

While the paper contains some interesting new results, it represents a largely incremental step over previous work. Moreover, the concept of "broadband" microcoils as presented in the manuscript is poorly motivated and lacks substance, as discussed in more detail below. For these reasons, I recommend rejection of the manuscript in its present form. Specifically, I have the following concerns and comments:

First, we would like to stress that the current paper is not about the broad band coil concept (viz. untuned microcoil circuits); this concept has been published in this journal (nature communications 2014; and patented) 9 years ago and has been reviewed by one of us in collaboration with an electrical engineer (Anders & Velders 2018) as well as by others (Davoodi et al. JMR2019 and SciRep2021), as already listed in the original ms. We believe our 1D & 2D trinuclear NMR data on such small volumes and amounts of material with just a single microcoil in an untuned circuit is unique and very much supporting the concept.

1. Compared to previously published work, the present manuscript introduces planar micro coils and a changed fluidic geometry. The authors claim that these planar coils have some special "broadband nature", due to which one can forgo the usual (complex) hardware needed for multiple-frequency tuning and matching. This assertion is repeated without evidence throughout the paper, implied by referring to "the broadband coil" etc. This does not, unfortunately, make it any more convincing. Microcoils do not defy the laws of electrodynamics just because they are small.

We believed that calling our set-up a broad-band microcoil, as a *pars pro toto* style figure, would allow, in particular also non-expert readers, to directly grasping the essence of our work, *i.e.* indeed executing multifrequency experiments without complex hardware. We notice the reviewer's problem with our broad-band microcoil definition (here in point 1, and also in point 3 and 4). We would like to avoid a semantic discussion, so we have adapted the term throughout the text in more cumbersome ways such as "untuned non-resonant coil" or "microcoil in a non-tuned RF circuit".

2. The microcoil terminates a 50 Ohm transmission line with an impedance that is likely to be dominated by an inductance, which by experience lies in the range of a few nH for a coil of this size. Together with a small resistive component (say, 0.1 Ohm), this translates to an impedance at 500 MHz of about $Z=(0.1+5j)\text{Ohm}$, extremely far from a matched condition for any of the frequencies in question. As a result, the entire setup, including the 50 Ohm transmission lines, filters, and connecting cables, will exhibit a very large standing wave ratio, leading to a large fraction of the signal power being dissipated in the connection network. At the same time, the signal from the coil does not benefit from resonant amplification. Under these conditions, the preamplifier noise (rather than the Johnson noise from the coil) dominates, resulting in a loss in sensitivity. This is clearly visible in the results presented in Fig 3: from the ^1H spectrum (Fig 3d) one can discern a signal/noiseratio of about 30. Together with the proton concentration ($2 \times 13.25\text{M}$) contributing to this signal, and a line width of about 10Hz, one obtains a number limit of detection (nLOD) of about 20 nMol \sqrt{s} . The authors present a more favourable LOD calculation from the results in Fig 5a; however, it ignores the fact that the methylene

multiplet is not resolved in the spectrum, and therefore is probably too optimistic. Detectors of similar volume have been presented in the literature with nLODs around 0.1 nMol sqrt(s) (e.g., van Bentum et al, J. Magn. Reson. 189, 104-113 (2007)).

The reviewer is largely underestimating the direct resistance of planar spiral microcoils, which in our case is in the order of 8 Ohm and not 0.1 Ohm, so almost two orders of magnitude bigger. The ohmic resistance of the microcoil therefore contributes significantly to the impedance. This by the way has also been published already in our earlier (Fratila et al. Nat. Commun. 2014) paper. The reviewer states many factors are not optimal for classical NMR detection; that is correct, and we never claim they are. The point the referee forgets to mention here (but does rightly so under point 4) is that despite these non-optimal conditions in the classical NMR circuit paradigm, the spectra themselves proof the paradoxical use of a non-tuned circuit to be very feasible, not only with simple 1D but even with complex 1D & 2D trinuclear experiments.

Regarding figure 3, the SNR we calculated was ~ 80 , and the $nLOD_{600,f}$ is then $7.4 \text{ nmol} \cdot \text{Hz}^{1/2}$, so three time better than the reviewer estimated.

Regarding the point of the multiplet overestimation, we do not exactly understand the question. We anyways repeated the calculation on a water signal to have a singlet, and obtain a $nLOD_{600,f}$ of about 1.0 nmol rather than 1.5 on the TFE signal, so well within the same range. We thank the reviewer and updated the main text and methods section accordingly

Finally, the reviewer refers to a paper (JMR 2007) in which a different way of calculation for the LOD is used, so complicating a direct comparison. In fact, this corroborates our reluctance in giving too much emphasis on the LODs, and instead directly use NMR spectra to compare performance.

3. The argument for the "broadband" nature of these coils needs to be substantiated, and the case for their use balanced against the tradeoff in sensitivity.

The argument for the broadband nature and the balance against tradeoff in sensitivity has been substantiated and discussed extensively in our earlier work (Nat. Commun. 2014) and by others (Davoodi et al. JMR2019 and SciRep2021), as already listed in the original ms.. The argument is simple and stated already by the referee self in the comment 4: *"There is merit in the simplicity of the approach, which contrasts with the complexity of coil tuning and matching, particularly if multiple frequencies are required."* Also other experts in the field have quantified the tradeoff. Loss of sensitivity is acceptable and stated by Korvink and co-workers to be $\sim 37\%$. Of a comparably sized tuned and matched resonator. We have added this in the experimental section to substantiate the discussion on trade off.

A direct comparison of a tuned versus non-tuned circuit is not trivial, as in a tuned circuit the quality depends on, e.g., quality of the capacitors used, the soldering and other connections, wire properties just to mention a view. With respect to a classical tank circuit, the untuned circuit is not ideal for SNR; Korvink et al. have come up with a 37% sensitivity compared to a tuned and matched circuit.[]

4. That said, the results presented by the authors in this and previous publications do show that in spite of this, NMR signals can be acquired successfully. There is merit in the simplicity of the approach, which contrasts with the complexity of coil tuning and matching, particularly if multiple frequencies are required. However, the implication that there is some magic property of these micro coils that makes them "broadband" is misleading at best if the severe tradeoff in sensitivity is not discussed alongside.

First of all, we are pleased to see the reviewer recognizing the value of the concept, whether it is called broadband or differently. Simplicity and affectivity is indeed clear. Having said that, there is no magic

involved, and we don't claim that either. Despite this simple set-up that goes against the classical paradigm of RF tank circuits, we manage to achieve even very complex trinuclear experiments, with just one single microcoil. Broad band is not misleading in our eyes, and is not used to that purpose. As said before, we have changed this into more cumbersome definitions such as *microcoil in non-tuned circuit*.

5. The present manuscript shows that multinuclear HMBC and HMQC experiments are feasible in this setup at thermal polarisation (without CIDNP) with highly concentrated samples (neat trifluoroethanol). While this is interesting as a demonstration of principle, it is unclear what the use case would be if such highly concentrated samples are necessary.

The TFE is indeed a good demonstration of principle. It is very common within the NMR specifications of commercial probes to use neat or very concentrated samples, or even isotope-enriched samples. Moreover, we use the same demonstration sample here as we used in the Nat. Commun. 2014 paper, to allow for a straightforward comparison. The merit of the TFE spectra for demonstration purposes is in particular lying in the skewed patterns due to the trinuclear scalar-coupling interactions in which the origin of the skewed pattern can be illustratively demonstrated. Regarding concentration sensitivity, we exploit and shown the combination with photo-CIDNP, allowing HF-HMQC on parafluorophenol at 13 mM concentration in figure 5d, which is far from being highly concentrated. We have updated figure and highlighted the peak of the internal reference of 300 mM clearly visible in the 1D & 2D spectra, and also the 13 mM parafluorophenol peak, emphasizing 2D info is obtained not only in neat TFE but also for samples in the sub-M (not polarized) and in the lower mM range (when polarized).

Updated Figure 5. Comparison of the photo-CIDNP broadband planar spiral setup with commercial 500 MHz probe and with a 1200 MHz NMR system. **A)** Single-scan 500 MHz ^1H spectrum of 325 picomole (13 mM) of *p*-fluorophenol, 7.5 nanomole (300 mM) of TFE and 28 picomole (1.1 mM) of FMN in H_2O ($n\text{LOD}_{f,600}$ 1.0 nmol $\cdot\text{Hz}^{1/2}$), in the planar spiral microcoil. In the right inset the TFE quartet is visible (yellow c), but *p*-fluorophenol (left inset) is not observed. **B)** 1200 MHz ^1H spectrum with the same amount of TFE and *p*-fluorophenol as in (a) dissolved in 200 μL D_2O on a 3mm cryoprobe. Also here, only the TFE quartet is observed after 1 scan ($n\text{LOD}_{f,600}$ 1.0 nmol $\cdot\text{Hz}^{1/2}$). **C)** ^{19}F photo-CIDNP spectrum of a 13 mM *pF* sample, where the 300 mM TFE signal (green, d) is observed as well as the *p*-fluorophenol signal (purple, a) observed because of the hyperpolarization of the ^{19}F nucleus in 1 scan ($n\text{LOD}_{f,600}$ 2.9 nmol $\cdot\text{Hz}^{1/2}$ for the thermally polarized and an $n\text{LOD}_{f,600}$ of 0.01 nmol $\cdot\text{Hz}^{1/2}$ for the hyperpolarized *pF*). **D)** 2D photo-CIDNP ^{19}F HMQC spectrum at 2 $\mu\text{L}/\text{min}$ of the 13 mM *pF* sample as in (c) using the broadband NMR chip. The ^{19}F - ^1H cross-peak of *p*-fluorophenol (a,b) as well as TFE's (c,d) are revealed in 32 scans. **E)** Same amount of sample and experiment (^{19}F - ^1H HMQC) as in (d), but without photo-CIDNP, on a 500 MHz Bruker NMR system equipped with an iProbe. A similar NMR acquisition time does not render the corresponding *p*-fluorophenol peak, only the TFE can be detected. Note that all photo-CIDNP experiments were carried out in continuous light irradiation mode, simplifying the typical 2D photo-

CIDNP pulse sequences⁴⁷, in combination with continuous flow regime to avoid loss of signal because of the accumulation of photodegraded photosensitizer in the detection volume.

6. The discussion of sensitivity of the CIDNP results (Fig 5) should be done in terms of clear indications of the nLOD figures for each case. The comparison of signal amplitudes obtained by averaging different numbers of transients and under different conditions (field strength, sample volume, coil size, with/without CIDNP amplification) makes it impossible to discern the true advantages of the approach.

We do not get the point from the reviewer; the 1Ds provided in figure 5 are all single scans. Moreover, the nLOD has been introduced by others (Iepucky et al., [67]) on purpose and specifically in an attempt to normalize data obtained in different settings and under different conditions. Further, although the nLOD has its use in estimating the comparability of different coils and probes, the real proof of the functioning and use of a coil/probe is in its NMR data. Also in standard NMR probes, the probes are typically optimized for a subset of experiments (for example ¹H only, ¹H, ¹³C, etc.) and manufacturers sell a multitude of probes, each for typically more than 50 kEuro. Probes with 2D compatibility, moreover, practically by definition, have compromised specific 1D properties. We repeat, with the direct comparison with the state-of-the-art commercial 500 MHz and 1200 MHz spectrometers, we provide people a direct comparison, which in our opinion says perhaps more than the abstract nLOD numbers. We anyways have added the nLOD_{6,600} values in the updated figure 5 caption (see here above).

7. (Related point) The results section claims "... photo-CIDNP on the broadband planar spiral coil ... opens up the possibility of measuring ... samples in the hundred(s) of picomole samples". This is misleading, as this level of sensitivity has been demonstrated by multiple authors at the same detector size WITHOUT hyperpolarization.

This section is about *heteronuclear 2D NMR*; so, to avoid misinterpretation the new sentence is (in line with the conclusions and message of the ms.):

... opens up the possibility of measuring advanced heteronuclear 2D NMR on samples in the hundred(s) of picomole range.

The 1D LOD of our set up is obviously significantly lower than for the 2D. We have actually reached the 0.5 mM sample of pF, and the nLOD_{6,600} is 0.01 nmole·VHz.

Minor points:

8. The manuscript pages should be numbered.

OK, done.

9. Fig 1: instead of numbering the traces, indicate the time (left panel) or the diode current (right panel). The scale indicator "Light intensity" with units of mA does not make sense, and is confusing given that the numbers next to it refer to the trace number, not the diode current.

OK, done.

10. Scheme 2 and Figure 2a would benefit from a size scale.

OK, Done.

11. Fig. 4: axis should be clearly labelled, e.g., "¹³C Chemical shift [ppm]" etc

OK, done.

12. Fig 5: see point 11

OK, done.

Reviewer #3 (Remarks to the Author):

The manuscript 'Multinuclear 1D & 2D NMR with ^{19}F -Photo-CIDNP hyperpolarization in a microfluidic chip with broadband microcoil' by Gomez and co-workers describes the integration of a new home-made device for sample illumination with the nanoliter-scale microfluidic broad-band NMR chip previously introduced, as well as the on-flow photo-CIDNP experiments performed to assess this device. This work is a continuation of previous developments of this team, dealing with broadband NMR micro-detection (ref. 8), microfluidics and photo-CIDNP (ref. 44), its aim being to combine microfluidic and photo-CIDNP approaches with broadband (trinuclear) NMR detection. The device is ingenious, well designed and reveals good performances in terms of hyperpolarization by photo-CIDNP (applicability of the system to high optical density solutions) and of $\{^1\text{H}\}^{19}\text{F}$ (- ^{13}C) NMR. In the former experiments the practicality of the device is used to optimize the target-photosensitizer ratio. In the latter, heteronuclear inverse experiments show the capability of the microcoil to transmit ^{13}C , ^{19}F and ^1H rf pulses within a pulse sequence without individual tuning (true for HMQC/HMBC, but how to perform a HSQC?). The bibliography appears to be complete, with few or no missing references, and there is enough details in the Methods part for the work to be reproduced. Even if according to me the manuscript does not contain major scientific or technological breakthrough, the full integration and performances of the device make that this deserves to be published in Nature Communications.

We thank the reviewer for the very positive feedback and summary of our work. Regarding if/how to perform a HSQC; that is actually doable as well, as also shown in our Nature Communications 2014 paper. For practical reasons we typically use HMQC rather than HSQC as the former is more robust. We have now newly recorded an HSQC and provided the spectrum in the SI.

Figure S16. 2D ^{19}F ^1H -HSQC NMR on neat TFE in stopped flow. The total experiment time is 80 minutes. The acquisition parameters are 4 as number of scans and 512 as number of increments.

Minor remarks:

- Beginning of the Results & Discussion part: in the sentence 'the performance of photo-CIDNP in spiral planar broad-band planar microcoils...' maybe remove one of the two 'planar'

OK, done

- Figure 1 is not very clear: 1) the intensity scale of the peaks should be increased (even if it means that the spectra overlap, which is not a problem since they are offset), 2) for the sake of clarity on the right-hand series the light intensity should directly constitute the y axis, 3) the last sentence in the caption should be removed and made part of the main text only, as it is confusing.

OK, done

- Caption of Fig. S2: please indicate the units (s, W, etc.)

OK, done

REVIEWERS' COMMENTS

Reviewer #1 (Remarks to the Author):

The authors have identified the area they use for their volume calculations. I am happy to accept the paper in its current form.

Reviewer #3 (Remarks to the Author):

The authors have responded satisfactorily to the comments I have made (however with the exception of the one regarding the caption of Figure S2). The manuscript deserves to be published in Nature Communications.

Nature Communications manuscript NCOMMS-22-39303A

We thank both reviewer 1 and 3 for their continuing efforts and feedback on the paper. Below we answer to their final comments.

REVIEWERS' COMMENTS

Reviewer #1 (Remarks to the Author):

The authors have identified the area they use for there volume calculations. I am happy to accept the paper in its current form.

We are glad to hear reviewer 2 is satisfied.

Reviewer #3 (Remarks to the Author):

The authors have responded satisfactorily to the comments I have made (however with the exception of the one regarding the caption of Figure S2). The manuscript deserves to be published in Nature Communications.

We are glad to hear reviewer 3 is satisfied with our changes. In addition, we have now updated also the figure caption of Figure S2.

In addition, we have updated the ms avoiding the term "broadband microcoil" and replacing that with "untuned microcoil", to avoid misunderstanding where the broadband aspects are deriving from.